# Synchronized mesenchymal cell polarization and differentiation shape the formation of the murine trachea and esophagus

Keishi Kishimoto [1,2], Masaru Tamura [3], Michiru Nishita[4], Yasuhiro Minami[4], Akira Yamaoka [1,2], Takaya Abe[2,5,6], Mayo Shigeta[2,5] & Mitsuru Morimoto [1,2]

Tube morphogenesis is essential for internal-organ development, yet the mechanisms regulating tube shape remain unknown. Here, we show that different mechanisms regulate the length and diameter of the murine trachea. First, we found that trachea development progresses via sequential elongation and expansion processes. This starts with a synchronized radial polarization of smooth muscle (SM) progenitor cells with inward Golgi-apparatus displacement regulates tube elongation, controlled by mesenchymal Wnt5a-Ror2 signaling. This radial polarization directs SM progenitor cell migration toward the epithelium, and the resulting subepithelial morphogenesis supports tube elongation to the anteroposterior axis. This radial polarization also regulates esophageal elongation. Subsequently, cartilage development helps expand the tube diameter, which drives epithelial-cell reshaping to determine the optimal lumen shape for efficient respiration. These findings suggest a strategy in which straight-organ tubulogenesis is driven by subepithelial cell polarization and ring cartilage development.

[1] Laboratory for Lung Development, RIKEN Center for Developmental Biology, Kobe 650-0047, Japan. [2] Center for Biosystems Dynamics Research, Kobe 650-0047, Japan. [3] RIKEN BioResource Center, Tsukuba, Ibaraki 305-0074, Japan. [4] Division of Cell Physiology, Department of Physiology and Cell Biology, Graduate School of Medicine, Kobe University, Kobe 650-0017, Japan. [5] Laboratory for Animal Resource Development, RIKEN Center for Life Science Technologies and Center for Biosystems Dynamics Research, Kobe 650-0047, Japan. [6] Laboratory for Genetic Engineering, RIKEN Center for Life Science Technologies and Center for Biosystems Dynamics Research, Kobe 650-0047, Japan. Correspondence and requests for materials should be addressed to M.M. (email: mitsuru.morimoto@riken.jp)

The delivery systems of multicellular organisms rely on the size and shape of tubular organs[1,2], and developmental disorders of tubular tissue cause congenital diseases in humans[3–5]. While organogenesis is progressed by growth factor-based epithelial–mesenchymal interactions, tubulogenesis studies have revealed that de novo luminal formation and the subsequent complex organization of small tubes, such as in mammary and salivary glands and the vascular system, are controlled by mechanical regulations of epithelial cells, involving skeletal structures and differences in cell–cell adhesiveness coordinated by synchronized cellular polarity[1,2,6–10]. However, the contribution of cell polarity in the surrounding mesenchymal cells to tubulogenesis is still unknown[11,12].

The trachea is the exclusive passage for delivering inhalation flow into the lung, which maintains the inhaled air by mucociliary clearance, humidification, and warming prior to entering the alveoli. The tube shape of the trachea determines ventilation efficiency[3,4,13]. The human trachea is about 13-cm long by 2-cm wide, which allows the passage of 30–120 l of air every minute. The mouse tracheal tube grows to 3-mm long and 500 μm in external diameter by E18.5 (Fig. 1a–d). This large and simple tube is composed of several tissue compartments: endoderm-derived pseudostratified columnar epithelium, and mesoderm-derived mesenchyme, including smooth muscle (SM), C-shape cartilage rings (Fig. 1a), vagal nerves, as well as blood vessels. Rigid cartilages support the ventral and lateral sides to maintain the tube shape, and SM tissue connects the cartilages at the dorsal side, which also provides elasticity[14]. Because ventilation efficiency is determined by the tube shape, developmental defects of the tracheal/lung lineage specification or cartilage formation contribute

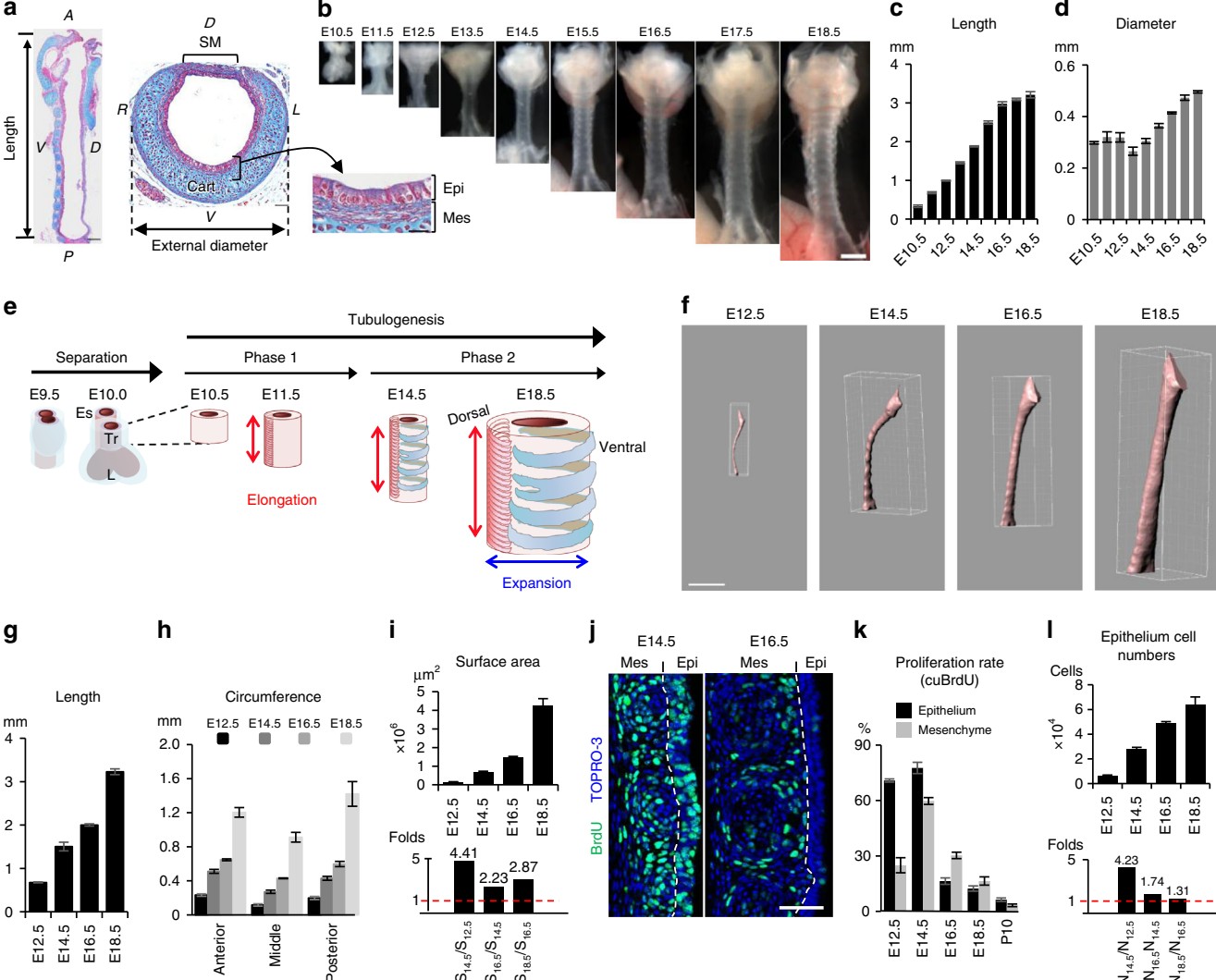

**Fig. 1** Tracheal tubulogenesis process. (**a**) Azan staining of trachea at E18.5. Length was defined as the distance from the larynx to the main branch. (**b**) Gross morphology of developing trachea. Tube length (**c**). External diameter (**d**). Data represent means ± SEM ($n \geq 3$). (**e**) Schematic model of trachea tubulogenesis. Red and blue arrows indicate length and diameter expansion. **f** 3D-luminal structures of whole tracheas reconstructed from micro-CT images using IMARIS. See also Supplementary Movie 1. Quantification of the length (**g**), circumference (**h**), and luminal-surface area (**i**) ($n \geq 4$). Change in luminal surface area (**i**, upper panel) and the fold change (**i**, lower panel) over time are shown. Data represent means ± SEM ($n \geq 3$). **j** Sections of BrdU-incorporated developing trachea were stained for BrdU (green) and TOPRO-3 (blue). Dotted lines indicate epithelium–mesenchyme boundary. **k** Rate of proliferating cells determined by cumulative BrdU-incorporation assay. Data represent means ± SEM ($n \geq 4$). **l** The numbers of epithelial cells (upper panel) and fold changes (lower panel) in whole trachea over time. Data represent means ± SEM ($n \geq 3$). A anterior; D dorsal; L left; P posterior; R right; V ventral; cart cartilage; SM smooth muscle; Epi epithelium; Mes mesenchyme; Tr trachea; Es esophagus. Scale bar = 500 μm (**b**, **f**), 200 μm (**a**; sagittal view), 50 μm (**a**; transversal view, **j**)

to serious pediatric diseases, such as tracheostenosis and tracheomalacia[3,4,15].

The embryonic tracheal/lung progenitors originate from the ventral foregut of the early embryo[15–17]. In the mouse embryo, trachea/lung development is initiated at E9.0 by the separation of these progenitors from the surrounding splanchnic mesoderm. The tracheal anlagen initially arises from the ventral foregut to form a diverticulum, coincident with the invagination of lung buds into splanchnic mesoderm, driven by growth factor-based epithelial–mesenchymal interactions[18]. The tracheal and lung bud epithelia form a tube structure that begins to elongate in a distal direction. Tracheal SM-cell differentiation begins at E11.5 within a restricted area of dorsal tracheal mesenchyme and forms circumferentially bundled structures. Chondrogenesis starts at E14.5 to support the ventral and lateral sides of the tracheal tube. However, the mechanisms underlying tracheal tubulogenesis after the initial separation are poorly understood.

Quantitative analyses of cell shape during tracheal morphogenesis of the fruit fly have elucidated mechanisms for tube length regulation[19,20]. In mammals, the branching lung buds have been completely mapped, and the genetic regulations of these complex and stereotyped airway structures have been determined[21]. Here, to discover the mechanisms underlying mammalian tracheal tubulogenesis, we perform comprehensive quantitative analyses of this process in mouse, from the cellular to whole-organ scale.

## Results

**The trachea undergoes sequential elongation and expansion processes.** We first measured the length and external diameter of developing tracheas (Fig. 1b–d). The tube length was continuously elongated along the anterior–posterior axis, whereas the external diameter was unchanged from E10.5–E14.5. After E14.5, the tube grew in both length and diameter. These observations and previous reports on trachea-esophagus separation[18,22] led us to hypothesize that, after the initial separation (E9.5–E10.5), the developing mouse trachea is shaped via two tubulogenesis steps: first, tube length elongation (E10.5–E14.5), and second, diameter expansion in addition to elongation (E14.5–E18.5) (Fig. 1e).

Although the initial separation is well studied, the subsequent tubulogenesis processes are not well understood. We, therefore, quantified the epithelial morphogenesis during tracheal tubulogenesis. The luminal-surface area of the developing trachea was measured by a combination of high-resolution micro-computed tomography (micro-CT) scanning and 3D-image analysis (Fig. 1f–i and Supplementary Movie 1). Consistent with the gross morphology measurement, the length constantly increased (Fig. 1g). Unlike the external diameter, the internal circumference was expanded throughout embryogenesis (Fig. 1h). The epithelial surface area was undergoing an accelerating expansion (Fig. 1i).

To investigate the mechanisms of tube growth and luminal enlargement, we next evaluated the proportion of proliferating cells in the developing trachea from E12.5 to E18.5 using a BrdU (5-bromo-2′-deoxyuridine) incorporation assay. The proportion of proliferating cells peaked at E14.5 in both mesenchyme (59.4%) and epithelium (77.7%) (Fig. 1j, k and Supplementary Fig. 1a). Detailed evaluation with another mitotic marker phospho-histone H3 revealed that the cell-proliferation rate peaked precisely at E15.0 (Supplementary Fig. 1b, c). These data revealed that the proliferation rate decreased after E15.0, while the tube diameter and luminal structure were expanding.

To accurately determine the increase in the epithelial-cell population during each time period ($N_{t1}/N_{t2}$), we counted the epithelial cells of the entire trachea at several developmental time points (Fig. 1l, upper panel). This counting was conducted by whole imaging of the tracheal epithelium at single-cell resolution

and image analysis using IMARIS. This analyses indicated the almost linear expansion of epithelial-cell numbers with modest deceleration (Fig. 1l). The epithelial-cell numbers increased 4.23-fold from E12.5 to E14.5 ($N_{14.5}/N_{12.5}$, Fig. 1l, lower panel), consistent with the luminal-surface increase (4.41-fold) measured by micro-CT (Fig. 1l, lower panel), demonstrating that cell proliferation was the major reason for the luminal-surface enlargement until E14.5. After E14.5, unlike the surface enlargement detected by micro-CT, the proliferation rates in both the epithelium and mesenchyme were progressively decreased (Fig. 1k, l). $N_{18.5}/N_{16.5}$ was only 1.31, whereas the surface area increased 2.87-fold, implying that epithelial-cell proliferation was not solely responsible for the luminal-surface enlargement after E14.5. We also observed that the cell-division orientation was random throughout development, suggesting that the directed tube elongation was not elicited by oriented cell division (Supplementary Fig. 2).

These analyses revealed that trachea tubulogenesis progresses via an "elongation" stage (depicted as Phase 1) followed by "expansion" (shown as Phase 2) (Fig. 1e). We further found that there is an accelerating nature of rate of surface area expansion.

**Luminal enlargement.** To investigate the mechanism underlying the luminal-surface enlargement that occurs with a low proliferation rate in trachea tubulogenesis phase 2, we examined the cell morphology and alignment of the developing epithelium in detail. At E18.5, the mature trachea has a pseudostratified columnar epithelial structure, which consists of a single layer of columnar luminal cells and flat basal cells (Fig. 1a). We found that from E14.5 to E18.5, the epithelial cells gradually changed shape to generate a typical pseudostratified columnar epithelium (Fig. 2a). To analyze this epithelial-cell reshaping quantitatively, we visualized cell membranes and nuclei three-dimensionally (3D) in $SHH^{Cre}$; $R26R^{RG}$ mice[23], which express membrane-GFP and histone-H2B-mCherry in the endodermal epithelium, and reconstructed a part of the epithelial structure on a PC. This 3D reconstruction revealed obvious epithelial-cell shape changes during phase 2 (Fig. 2b and Supplementary Movie 2). At E14.5 and E16.5, about 60% of the total population were luminal cells whose apical surface was exposed to the lumen, and 40% were basal-side cells that did not have an apical surface (Fig. 2b, d), and cells of various shapes were packed within a small space. From E16.5 to E18.5, the apical-surface area increased 1.5-fold, and the proportion of luminal cells increased to 80% (Fig. 2b, c), indicating that several basal-side cells had acquired an apical surface. These observations revealed that both "apical enlargement" and "apical emergence" contributed to the luminal-surface enlargement in addition to modest cell proliferation. Integration of these values estimated a 2.90-fold luminal-surface enlargement, due to increased cell numbers (1.31-fold), epithelial-cell reshaping including apical enlargement (1.55-fold), and apical emergence (1.43-fold) (Fig. 2e). This integrated value was almost equal to the luminal-surface enlargement quantified by micro-CT (2.87) (Fig. 1i). Thus, these three events were sufficient to explain the luminal area enlargement occurring from E16.5 to E18.5. To assess the impact of excess epithelial-cell proliferation in phase 2, we generated $Nkx2.1^{CreERt2}$; $LSL-Kras^{G12D}$ mice and induced excess proliferation by injecting tamoxifen for 3 days from E14.5 (Supplementary Fig. 3a–f). Tamoxifen injection increased the phospho-ERK1/2, as a downstream effector of Ras, and the expression of the mitotic marker Ki67, indicating that excessive epithelial-cell proliferation occurred in the trachea of the transgenic mice (Supplementary Fig. 3d–f). At E18.5, the Kras-activated epithelium exhibited an altered pseudostratified columnar epithelial structure, accompanied by failed apical

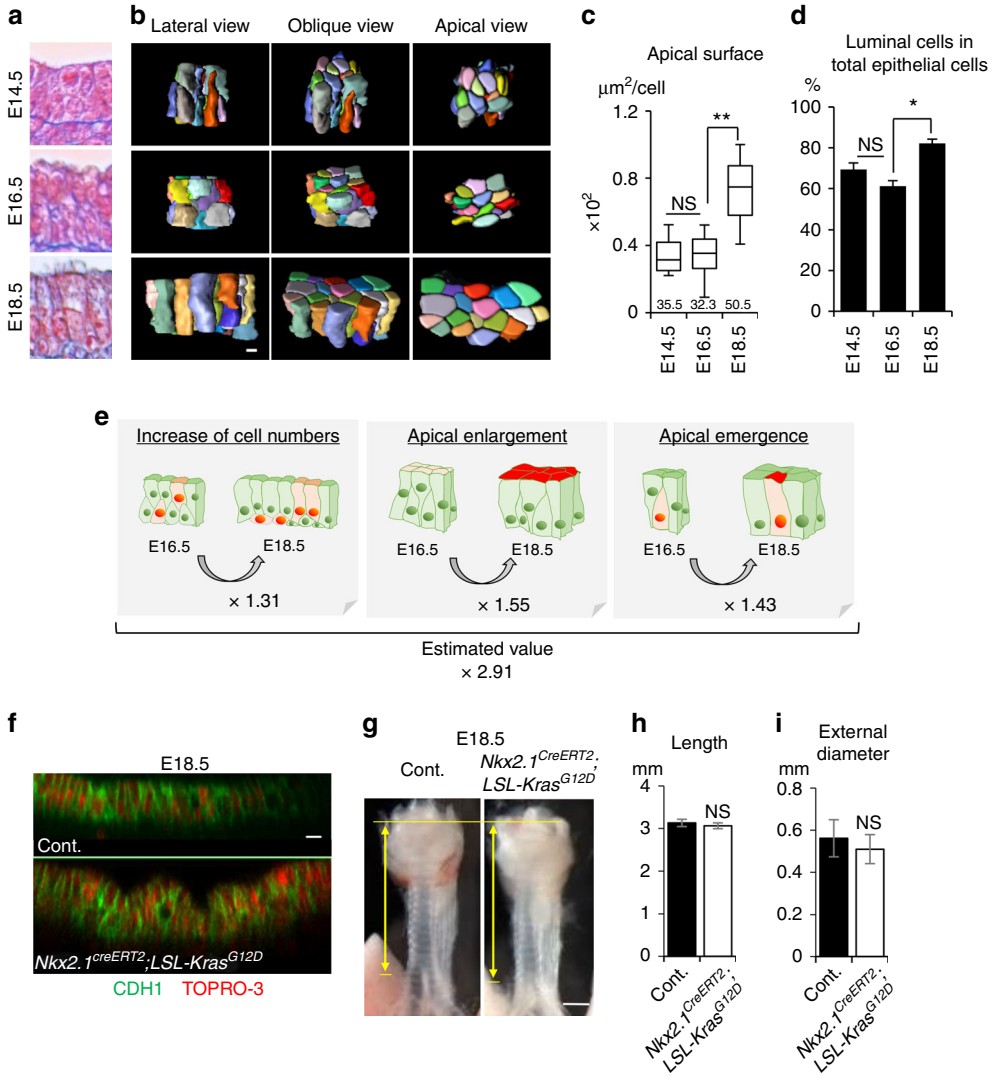

**Fig. 2** Low-proliferative luminal-surface enlargement with epithelial rearrangement after E14.5. **a** Azan staining of developing trachea epithelium. **b** Representative 3D reconstructed images of an epithelial-cell cluster (25 cells). See also Supplementary Movie 2. **c** Apical-surface area. Numbers represent means. ($n = 217$, 253, and 189 in three embryos for E14.5, E16.5, and E18.5, respectively). center line, box limits, and whiskers represent mean, 25 and 75% confidence limits, and min and max values, respectively. **d** Proportion of luminal cells in total epithelial cells. Data represent means ± SEM ($n \geq 3$) ($n = 528$, 732, and 1045 in three embryos for E14.5, E16.5, and E18.5, respectively). **e** Integration of quantitative values revealed that modest increases in cell numbers, apical enlargement, and apical emergence contributed to luminal-surface enlargement. **f** Optical image of tracheal epithelium in $Nkx2.1^{CreERt2}$; $LSLKras^{G12D}$ trachea and littermate control at E18.5. Whole tracheas were stained for CDH1 (green) and TOPRO-3 (red). **g** Gross morphology of $Nkx2.1^{CreERt2}$; $LSL-Kras^{G12D}$ trachea and littermate control. Yellow arrows indicate trachea length. Tracheal tube length (**h**). External diameter (**i**). Data represent means ± SEM ($n \geq 3$). $P$ values ($^{**}P < 0.01$, $^{*}P < 0.05$) show the significance with the Tukey-Kramer method (**c**, **d**) or Student's $t$ test (**h**, **i**). NS not significant. Scale bar = 500 μm (**g**), 20 μm (**f**), 5 μm (**b**)

enlargement and apical emergence, while the tube shape was intact (Fig. 2f–i, Supplementary Fig. 3a–c, g, h). These data suggested that the regulation of epithelial proliferation during a particular time window is crucial for the epithelial-cell reshaping that forms the pseudostratified columnar epithelium but not for tubulogenesis. In contrast to a previous report showing that an epithelial property determines tube morphology[1,2,7–9], our results suggest that the surrounding mesenchyme restricts epithelial expansion to shape the trachea.

**Wnt5a-Ror2 signaling extends the tube.** To determine the mechanism of tube elongation in tubulogenesis phase 1, we examined *Wnt5a*-knockout mice, which have a short-trachea phenotype[24]. We collected the tracheas of $Wnt5a^{-/-}$ mice during

development. We found that the tracheal elongation of $Wnt5a^{-/-}$ was defective from E11.5, and reached only 60% of the normal length at E18.5 (Fig. 3a, b). The $Wnt5a^{-/-}$ trachea showed a similar proliferation rate in the epithelium and mesenchyme, and a similar epithelial-cell number per area, as the control (Supplementary Fig. 4a–c). The number of cartilage rings decreased from $13 \pm 1$ to $6 \pm 1$, while their shape and periodicity were retained (Fig. 3a, Supplementary Fig. 4d). A delay in tube elongation appeared at E11.5 and the tube remained shorter than normal thereafter (Fig. 3b), suggesting that Wnt5a regulates the trachea length in phase 1. We, therefore, examined the time and pattern of *Wnt5a* mRNA expression by in situ hybridization. Intense *Wnt5a* expression was detected in the splanchnic mesenchyme surrounding the trachea epithelium at E10.5 and E11.5 (Fig. 3c),

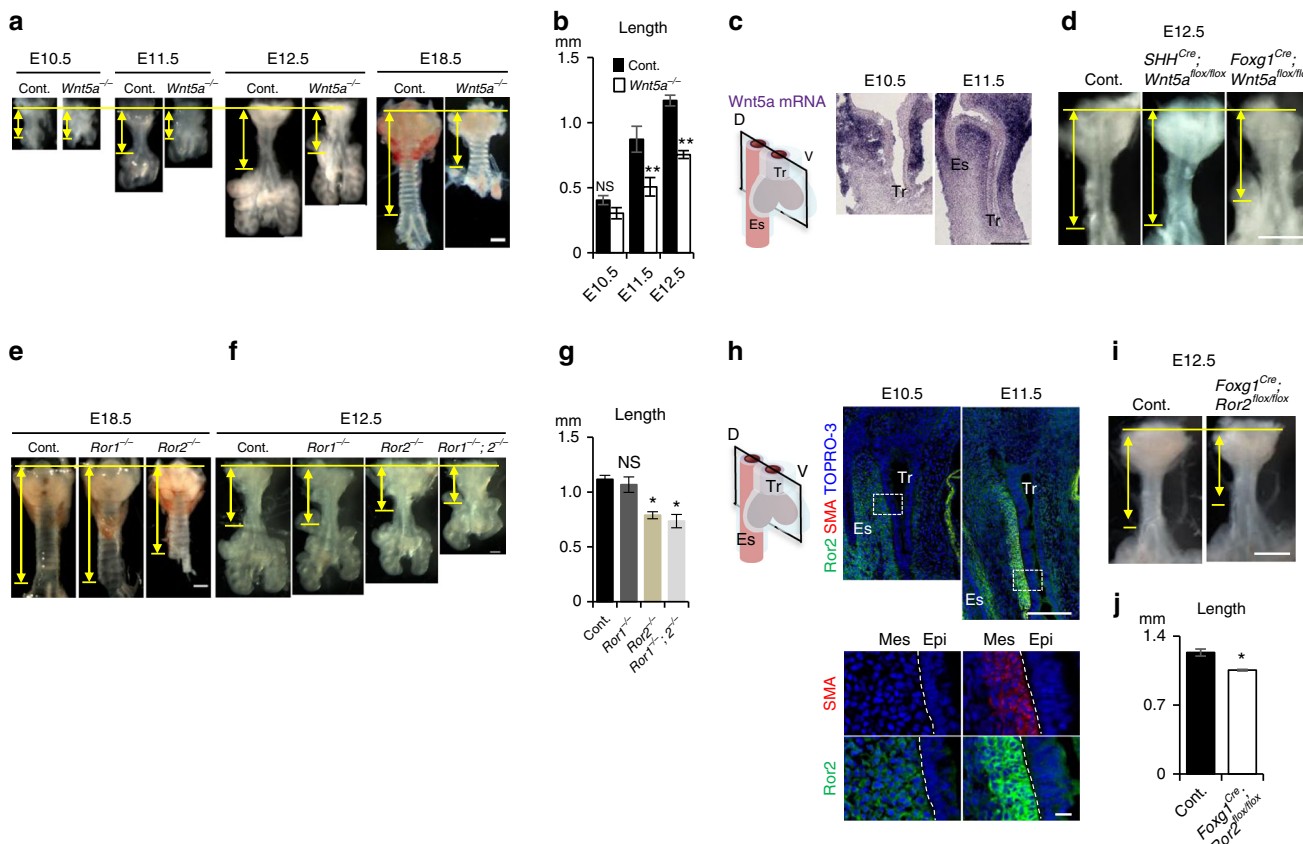

**Fig. 3** Wnt5a-Ror2 signaling is involved in tube elongation. **a** Gross morphology of a developing $Wnt5a^{-/-}$ trachea and littermate control. **b** Tracheal tube length (control; $n = 4$–6, $Wnt5a^{-/-}$; $n = 3$). **c** In situ hybridization for $Wnt5a$ mRNA in developing tracheas. **d** Gross morphology of $Foxg1^{Cre}$; $Wnt5a^{flox/flox}$ trachea ($n = 4$), $SHH^{Cre}$; $Wnt5a^{flox/flox}$ ($n = 3$) trachea, and littermate control ($n = 7$) at E12.5. Gross morphology of $Ror1$ and/or $2^{-/-}$ trachea and littermate control at E18.5 (**e**) and 12.5 (**f**). **g** Tracheal tube length (control; $n = 4$, $Ror1^{-/-} = 5$, $Ror2^{-/-} = 5$, $Ror1^{-/-}$; $Ror2^{-/-} = 3$). **h** Ror2 and SMA expression in developing tracheas. Sections were stained for Ror2 (green), SMA (red), and TOPRO-3 (blue). Lower panels show higher magnification images of dotted squares. Dotted lines indicate epithelium–mesenchyme boundary. **i** Gross morphology of $Foxg1^{Cre}$; $Ror2^{flox/flox}$ trachea and littermate control at E12.5. **j** Tracheal tube length (control; $n = 5$, $Foxg1^{Cre}$, $Ror2^{flox/flox}$; $n = 9$). Yellow arrows indicate trachea length. D dorsal; V ventral; Epi epithelium; Mes mesenchyme; Tr trachea; Es esophagus. $P$ values ($^*P < 0.05$, $^{**}P < 0.01$) show the significance with Student's $t$ test (**b**) or the Tukey-Kramer method (**g**). Scale bar = 500 μm (**a**, **d**, **e**, **i**), 200 μm (**c**, **e**, **h**; upper panel), 10 μm (**h**; lower panel)

indicating that Wnt5a is secreted from mesenchymal tissue during tube elongation. Next, we conditionally deleted Wnt5a from the splanchnic mesoderm, and definitive endoderm by generating $Foxg1^{Cre}$; $Wnt5a^{flox/flox}$ and $SHH^{Cre}$; $Wnt5a^{flox/flox}$ mice, respectively. The mesodermal mutant, but not the endodermal mutant, showed the short-trachea phenotype while the body appearance was indistinguishable from control (Fig. 3d, e and Supplementary Fig. 5a, b). Heterozygotes and homozygotes of the $Foxg1^{Cre}$ allele did not exhibit gross changes in the tracheal morphology (Supplementary Fig. 5c–f). Although $Foxg1^{Cre}$ targets a minor epithelial population[25], the epithelial Wnt5a also did not appear to have a role in tube elongation. We further deleted Wnt5a using another mesenchyme-specific Cre-line, $Dermo1^{Cre}$. The $Dermo1^{Cre}$, $Wnt5a^{flox/flox}$ mice showed a short-trachea phenotype similar to that of $Foxg1^{Cre}$, $Wnt5a^{flox/flox}$ mice (Supplementary Fig. 5k, l). These data indicated that the mesenchymal Wnt5a functions in tube elongation.

We next sought to identify the receptor for the mesenchymal Wnt5a involved in tracheal elongation. Wnt5a activates non-canonical Wnt signaling via $Ror1/2$ receptors to regulate cellular functions, including cell polarity[26]. We found that $Ror2$, but not $Ror1$, deletion resulted in the short-trachea phenotype at E18.5 (Fig. 3e). $Ror1/2$ double deletion resulted in a more severe phenotype than $Ror2^{-/-}$ at E12.5, and was a phenocopy of

$Wnt5a^{-/-}$ (Fig. 3f, g), indicating that the trachea tube elongation in phase 1 is mainly regulated by Wnt5a-Ror2 signaling and that Ror1 might compensate for the loss of Ror2 expression. To identify the Ror2-expressing cells responsible for tube elongation, we performed immunostaining of the developing trachea. A Ror2-expressing subpopulation was identified in the dorsal tracheal mesenchyme, spatially restricted to areas adjacent to tracheal or esophageal epithelium (Fig. 3h). These cells were exclusive to SM tissue, and the Ror2 expression intensity increased as SM tissue developed. Conditional deletion of $Ror2$ from splanchnic mesoderm using $Foxg1^{Cre}$; $Ror2^{flox/flox}$ mice, but not from epithelium using $SHH^{Cre}$; $Ror2^{flox/flox}$, resulted in a shortened trachea (Fig. 3i, j, Supplementary Fig. 5g, h), demonstrating that the Ror2-expressing mesenchymal cells are involved in tube elongation. These results also indicated that the contribution of epithelial Wnt5a-Ror2 signaling is negligible. Similar to $Foxg1^{Cre}$; $Wnt5a^{flox/flox}$, these $Foxg1^{Cre}$; $Ror2^{flox/flox}$ embryos had a normal body size and shape (Supplementary Fig. 5i, j), excluding the possibility that an abnormal body shape of $Wnt5a$ or $Ror2$ knockout embryos indirectly disrupted their trachea development.

**Radial polarization of subepithelial cells arrange SM morphology.** To study the role of Wnt5a-Ror2 signaling in SM

development, we first examined the SM-cell proliferation by Ki67 staining. The SM-cell proliferation was not affected in the $Wnt5a^{-/-}$ trachea (Supplementary Fig. 6a, b). Next, we assessed whether the SM tissue morphology was altered in the $Wnt5a^{-/-}$ trachea. We imaged 3D SM tissue at single-cell resolution by

reconstructing multiple Z-stack confocal images of normal or $Wnt5a^{-/-}$ trachea (Fig. 4a). The shapes of individual SM cells were visualized by tracing the cell outlines (Fig. 4b and Supplementary Movie 3). In normal trachea at E12.5, the SM tissue appeared at the dorsal side of the tracheal mesenchyme as a

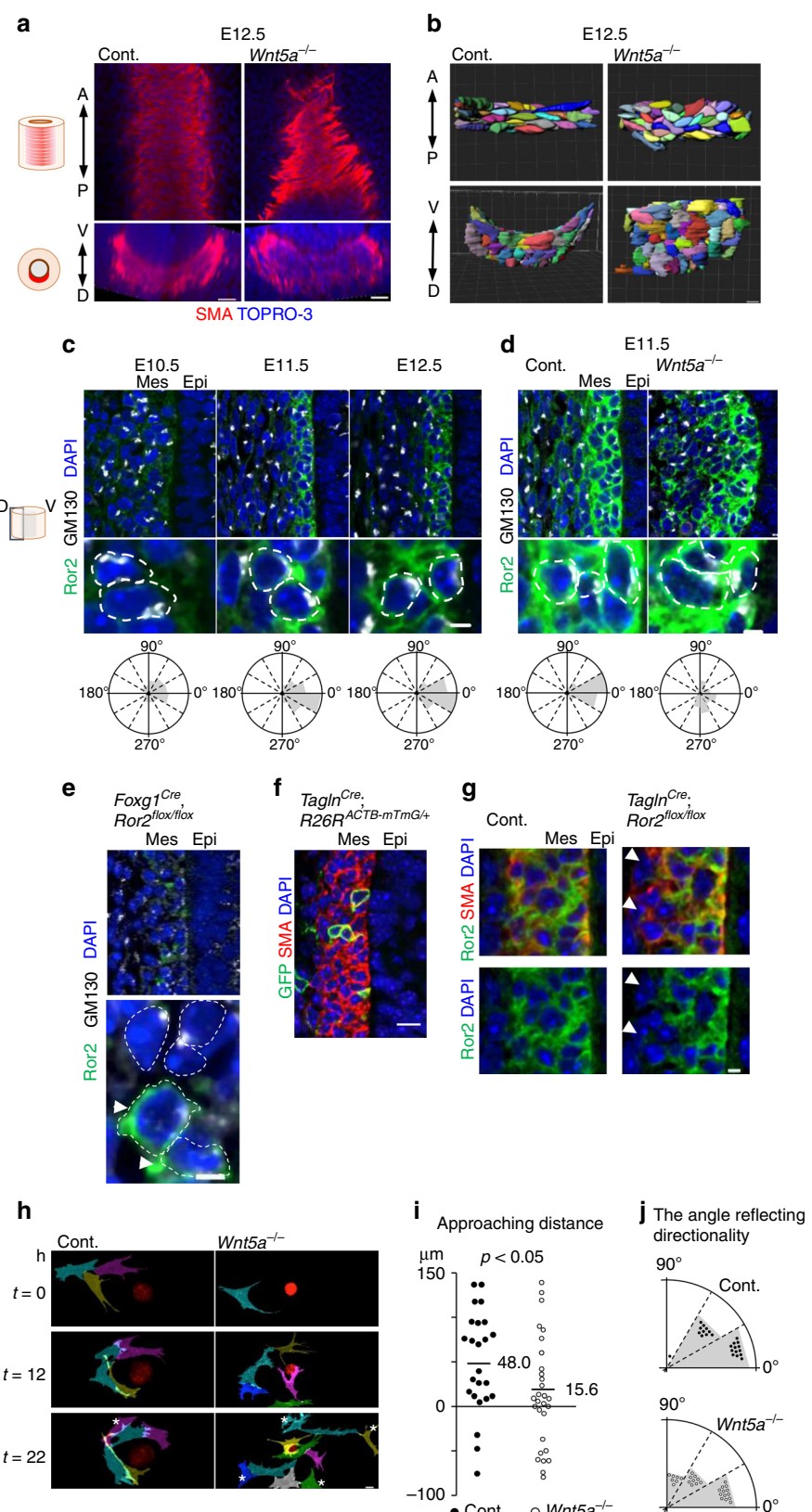

convex sheet along the epithelium. The SM cells were extended circumferentially and arranged in parallel to each other to form bundled structures. In $Wnt5a^{-/-}$ tracheas, although the SM differentiation was unaffected, the SM-cell alignments were altered and the SM tissue structure became flat and thick, suggesting that Wnt5a signaling is necessary for the SM-cell shaping and parallel arrangement.

We hypothesized that Wnt5a-Ror2 signaling aligns the SM cells and orients them circumferentially by coordinating polarized cell behaviors of the mesenchymal cells, analogous to the way the planer cell polarity of epithelial cells synchronizes epithelial-cell orientation. Due to the radial layer structure of the SM under the epithelium, we speculated that the SM cells were radially polarized toward the epithelium. We thus examined the polarization of the SM cells by determining the position of the Golgi apparatus and centrosome relative to the nucleus in individual Ror2-positive cells. Interestingly, the Golgi-apparatus and centrosome in SM cells were orientated toward the epithelium at E11.5 and E12.5 (Fig. 4c and Supplementary Fig. 7a). The polarity directions were synchronized and largely perpendicular to the epithelial sheet. The most of tracheal subepithelial cells showed this inward Golgi-apparatus displacement, indicating a radial pattern of SM-cell polarity (Supplementary Fig. 7b). This radial cell polarization was observed within the first three layers of subepithelial cells. Because the polarity was not synchronized at E10.5, before SM-cell differentiation, the radial cell polarization may begin at the onset of the fate determination into SM cells. Wnt5a deletion altered the synchronized cell polarity and deformed the layered SM tissue architecture (Fig. 4a, b, d), indicating that Wnt5a-Ror2 signaling is necessary for synchronizing the subepithelial cell polarization toward the epithelium to coordinate the radial cell polarity and build the proper SM architecture.

To understand how radial cell polarization contributes to SM morphogenesis, we conditionally deleted Ror2 within the splanchnic mesoderm. We confirmed that Ror2-negative SM cells exhibited random polarization, while some Ror2-positive tracheal SM cells resulting from incomplete targeting by $Foxg1^{Cre}$ displayed the proper radial cell polarity (Fig. 4e arrowheads). These observations suggested that Wnt5a-Ror2 signaling plays a cell-autonomous role in the radial cell polarization. In addition, a mosaic deletion of Ror2 was achieved by the inefficient gene-targeting of SM cells using the $Tagln^{Cre}$ line (Fig. 4f). While most of the Ror2-positive cells became integrated SM tissue, a few Ror2-negative SM cells were excluded from the subepithelial SM (Fig. 4g, arrowheads). These observations suggested that Wnt5a-Ror2 signaling enables SM cells to intercalate into the subepithelium. Therefore, to examine the migration of SM cells toward the epithelium directly, we performed live imaging of migrating SM cells in a 3D cell culture system (Supplementary Fig. 8). The developing trachea of $Tagln^{Cre}$, $R26R^{mTmG}$, in which

epithelial cells were labeled red and SM cells were labeled green, was isolated at E12.5 and digested with trypsin to single cells. These cells were then cultured on a glass-bottom dish coated with Matrigel to generate epithelial spheres. The migrating SM cells were observed by inverted multiphoton microscopy (Fig. 4h and Supplementary Movie 4). Normal SM cells within a short distance (200 μm) from an epithelial sphere migrated toward the sphere. In contrast, many Wnt5a-null SM cells maintained their distance from the epithelial spheres, and exhibited random migration (Fig. 4i, j). These observations indicated that Wnt5a is required for directional SM-cell migration toward the epithelium, which could be involved in tracheal tube elongation.

We reasoned that circumferentially connected subepithelial SM cells restrict the radial expansion and promote tube elongation in longitudinal axis[27]. To investigate this possibility, we examined the function of the tracheal SM. We examined airway peristalsis to observe SM contractility, which reflects the contractility of tracheal SM, in ex vivo culture[28]. Developing lungs were harvested from E11.5 embryos and cultured in the air–liquid interface fashion. Live imaging showed stochastic peristalsis of the trachea in control mice (Fig. 5a–c and Supplementary Movie 5). The $Wnt5a^{-/-}$ tracheas showed impaired contractility defined by the contractile rate, although the contractile frequency defined by the contraction and relaxation times was not altered. Thus, the SM in the $Wnt5a^{-/-}$ trachea showed an aberrant contraction. Although we do not think the periodicity is necessary for tube elongation, the static restriction by polarized SM tissue may constrain the epithelial tissue expansion and direct tube elongation along the anterior–posterior axis. Supporting this idea, Wnt5a-knockout or Ror2-knockout tracheas showed an expanded tube diameter and thick mesenchymal tissue (Fig. 5d, e, f). We further generated the SM tissue null trachea by deleting $Serum\ Response\ Factor$ ($Srf$) gene, a crucial transcriptional factor of SM-cell differentiation, using the $Foxg1^{Cre}$ line (Supplementary Fig. 9a). The SM-null trachea still retained the radial cell polarity (Supplementary Fig. 9b), suggesting that Wnt5a-Ror2 signaling coordinates the pattering and morphology of the SM progenitor cells via the radial cell polarity prior to SM differentiation at subepithelium. For our surprise, this mutant showed normal tube elongation and subepithelial cells showed circumferential supracellular-continuity of cell morphology likewise those of the normal trachea (Supplementary Fig. 9c, d). Furthermore, the axis of this supracellular connection was randomized in $Wnt5a^{-/-}$ trachea (Supplementary Fig. 9d). These data set imply that Wnt5a-Ror2 signaling plays a key role for forming circumferential cell–cell connection of SM progenitor cells at subepithelium by modulating the radial cell polarity.

We also observed radial cell polarization in the esophageal subepithelial cells (Fig. 5i). As seen in the trachea, in the $Wnt5a^{-/-}$ esophagus, both the synchronized subepithelial cell polarity and morphology of SM were altered, resulting in

---

**Fig. 4** Wnt5a-Ror2 signaling regulates SM morphogenesis by synchronizing the radial SM-cell polarity. **a** SM morphology in $Wnt5a^{-/-}$ trachea and control at E12.5. Whole tracheas were stained for SMA (red), TOPRO-3 (blue). Coronal (upper panel) and transversal (lower panel) sections. **b** Array of SM cells. Apical view (upper panel) and lateral view (lower panel). See also Supplementary Movie 3. **c, d** Polarity of Ror2-positive SM cells, determined by the Golgi-apparatus position relative to the nucleus. Sections were stained for GM130 (white), Ror2 (green), and DAPI (blue). White-dotted lines show the contours of cell membranes. Quantifications in three embryos are shown below. **c** Developmental time-course analysis in wild type (E10.5; $n = 115$, E11.5, $n = 214$, E12.5; $n = 236$). **d** In $Wnt5a^{-/-}$ trachea at E11.5 (control; $n = 99$, $Wnt5a^{-/-}$; $n = 73$). **e** Polarity of Ror2-negative or positive (arrowheads) SM cells in $Foxg1^{Cre}$; $Ror2^{flox/flox}$ trachea at E11.5: the lower panel shows magnified images of the upper panel. **f, g** Mosaic targeting of SM cells using $Tagln1^{Cre}$ line. GFP (green), SMA (red), DAPI (blue) in control (**f**) and $Ror2^{flox/flox}$ (**g**). Arrowheads indicate Ror2-negative SM cells. **h** Time-lapse sequence of epithelial sphere (red) and SM cells (magenta, cyan, gray, yellow, blue, and green) in Matrigel. Cells were labeled by Image J. *Cells keeping at a distance from spheres. See also Supplementary Movie 4. **i** Approaching distance of SM cells to sphere in **h**. Numbers represent means (control; $n = 23$, $Wnt5a^{-/-}$; $n = 29$). **j** Distribution of angles reflecting directionality to sphere in **h**. Scale bar = 100 μm (**e**, left), 10 μm (**a–c**; upper, **d**; upper, **e**; right upper, **g**; upper, **h**), 3 μm (**c**; lower, **d**; lower, **e**; lower, **g**; lower panels)

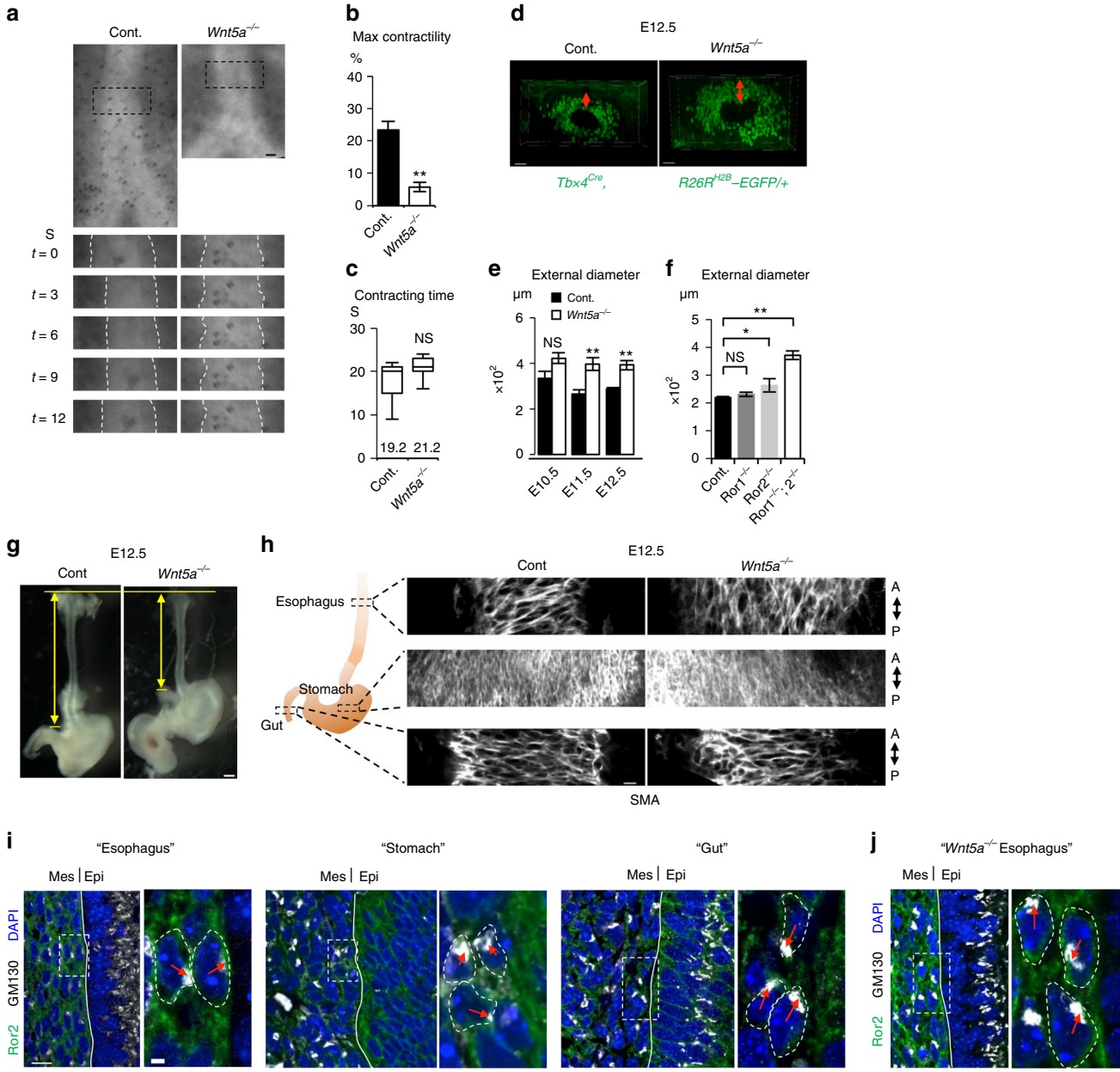

**Fig. 5** Wnt5a-Ror2 signaling controls contractility of SM and restricts radial expansion during tube elongation of developing trachea and esophagus. **a** Representative time-lapse sequence of contractile movement of tracheas. The black-dotted squares in upper are magnified in lower. White-dotted lines indicate epithelium–mesenchyme boundaries. See also Supplementary Movie 5. Maximal contractility (**b**) and contraction time (**c**) in **a**. Columns represent means ± SEM ($n \geq 3$). Numbers represent means. For box and whisker plot, center line, box limits and whiskers represent mean, 25 and 75% confidence limits, and min and max values, respectively. (**d**) Thickness of tracheal mesenchyme in $Tbx4^{Cre}$, $R26R^{H2B\text{-}EGFP/+}$. Transversal views were shown. Red-arrows indicate thickness of mesenchyme. (**e**, **f**). External diameter of tracheal tube. (Control; $n = 4$–6, $Wnt5a^{-/-}$; $n = 3$, $Ror1^{-/-} = 5$, $Ror2^{-/-} = 5$, $Ror1^{-/-}$; $Ror2^{-/-} = 3$). Each column represents the means ± SEM. $P$ values ($^{*}P < 0.05$, $^{**}P < 0.01$) show the significance with Student's $t$ test (**b**, **e**) or the Tukey-Kramer method **f**. NS not significant. (**g**) Gross morphology of the esophagus at E12.5. Yellow arrows indicate the length of the esophagus. **h** SM tissue architecture of digestive organs at E12.5. Whole organs were stained for SMA (white). Coronal sections were shown. A anterior, P posterior. See also Supplementary Movie 6. **i**, **j** The polarity of subepithelial cells of digestive organs from wild type (**i**) and $Wnt5a^{-/-}$ (**j**). Ror2 (green), GM130 (white), and DAPI (blue). Dotty squares were magnified in right panels. Lines indicate boundaries between epithelium and mesenchyme. Red arrow indicates the position of Golgi-apparatus relative to nuclei. Scale bar; 50 μm (**d**), 20 μm (**h**), 10 μm (**e**, **i**, **j**; left), 2 μm (**i**, **j**; right panels)

a short-tube phenotype (Fig. 5g and Supplementary Movie 6). These observations suggested that the mechanism in which Wnt5a-Ror2 signaling promotes long, straight tube morphogenesis is conserved. In contrast, in the gut, the mesenchyme exhibited randomized polarization in normal embryos, and $Wnt5a^{-/-}$ embryos showed no SM morphology phenotype (Fig. 5h). It is reported that epithelial-cell intercalation

mediated by Wnt5a-Ror2 is critical for tube elongation of the stomach, duodenum, and intestine[29,30]. We confirmed that Ror2 was expressed in the epithelium of the stomach and gut but not in the trachea or esophagus (Supplementary Fig. 10). Thus, different mechanisms are likely to be involved in the tube elongation in the trachea and esophagus vs. the stomach and gut.

**Cartilage growth expands tube diameter.** Next, we investigated the mechanism of tube expansion in tubulogenesis phase 2, which includes a resistance against the subepithelial restriction. Cartilage rings appear juxtaposed with SM to completely surround the tracheal tube[14]. Developmental cartilage disorders cause tracheal stenosis, because tracheal cartilage rings maintain the trachea's diameter[3,4,31]. We disturbed tracheal cartilage development by generating $Foxg1^{Cre}$; $Sox9^{flox/flox}$ mice, in which the splanchnic mesoderm lacks cartilage progenitors[14]. The mutant trachea developed without cartilage and was obviously narrowed by E18.5, whereas SM development and the tube length were not obviously changed (Fig. 6a–c, Supplementary Fig. 11). The lumen of the no-cartilage trachea became stenosed with a folding epithelial structure (Fig. 6d) due to the loss of mesenchymal rigidity. The total surface area of the no-cartilage trachea was measured by a method combining micro-CT and digital reconstruction. The mutant showed a smaller total luminal area that was 49.7% that of wild type. Notably, the mutant trachea developed normally until E14.5 (Fig. 6a–d), the onset of diameter expansion[32] (Fig. 1d). These results indicated that the tube expansion in phase 2 is accomplished by proper cartilage development after E14.5.

Finally, we examined the driving force of epithelial-cell reshaping observed from E16.5 to E18.5. We reasoned that the epithelial-cell reshaping is supported by the surrounding cartilage rather than by epithelial intracellular machinery. To evaluate the contribution of mesenchymal architecture to epithelial-cell reshaping, we examined the epithelial structure of the no-cartilage tracheas. We performed comprehensive epithelial-cell quantification, as in Fig. 2b–d, for the no-cartilage tracheas. At E18.5, both apical enlargement and apical emergence were inhibited in the mutant trachea (Fig. 6e–g and Supplementary Movie 7), whereas cell proliferation was unaffected (Supplementary Fig. 12). To characterize the epithelial-cell reshaping phenotype, we defined a "pseudostratification index," as the distribution of the distance from the nuclear centroid to the basement membrane[33] (Supplementary Fig. 13a). In wild type, the index started high, and decreased as epithelium developed (Supplementary Fig. 13b–c), reflecting epithelial-cell reshaping and pseudostratified columnar epithelium development. The mutant trachea, however, showed a higher index than the control at E18.5 (Fig. 6h, Supplementary Fig. 13d). To exclude the possibility that the Sox9-expressing mesenchyme sends signals to the epithelium to regulate apical rearrangement, we genetically ablated the *alpha 1(II) collagen* (Col2a1) gene, which is crucial for chondrogenesis under Sox9, using CRISPR-Cas9 genome editing (Supplementary Fig. 14a–c). The $Col2a1^{-/-}$ tracheas exhibited the narrowed trachea phenotype and failed to undergo epithelial reshaping, as seen in $Foxg1^{Cre}$; $Sox9^{flox/flox}$, while the Sox9-expressing chondrocytes were still present (Supplementary Fig. 14d–j). We also confirmed that $Col2a1^{-/-}$ did not affect epithelial-cell proliferation or differentiation (Supplementary Fig. 14k–n). Given that the loss of Sox9 reduces the basal cells and increase the club cell population[14,34], we conclude that the rigidity of the cartilage contributes to the diameter expansion and

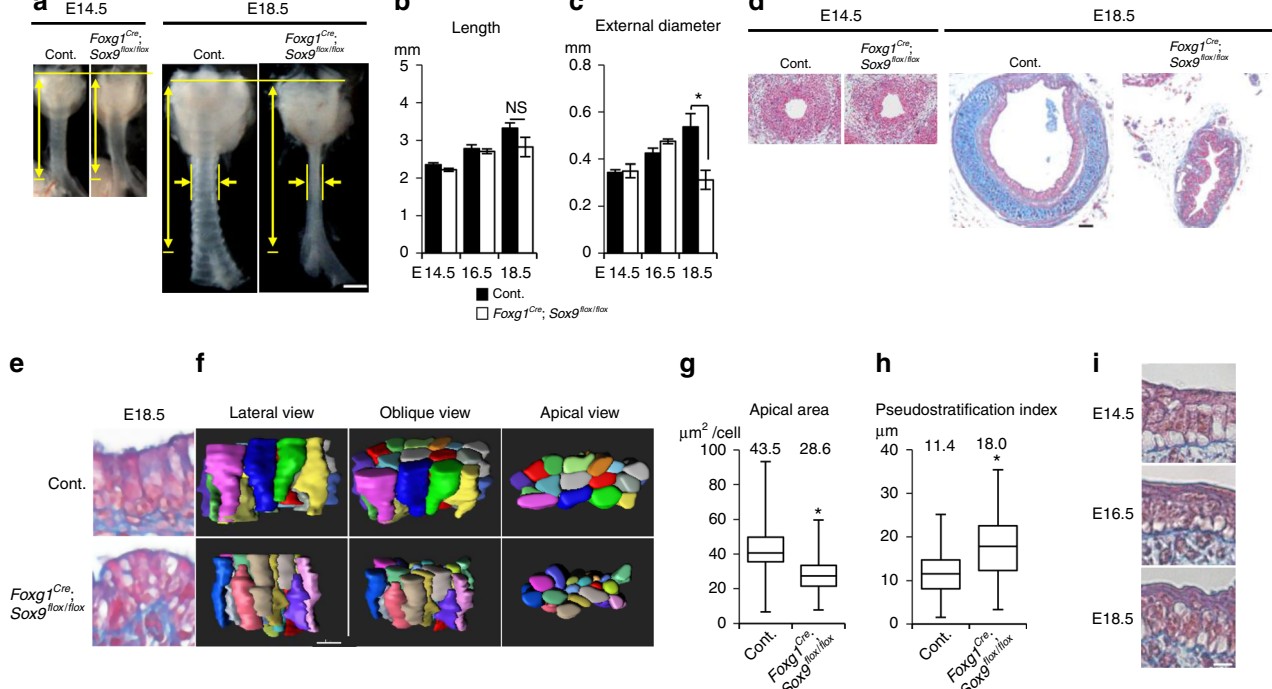

**Fig. 6** Cartilage development expands the tracheal but not esophageal diameter and promotes epithelial rearrangement for luminal enlargement. **a** Gross morphology of $Foxg1^{Cre}$; $Sox9^{flox/flox}$ trachea and littermate control. Yellow arrows indicate length and diameter of the tracheas. Measurement of tracheal tube length (**b**) and external diameter (**c**). (Control; n = 5–8, $Foxg1^{Cre}$; $Sox9^{flox/flox}$; n = 3 or 4). **d** Azan staining of transversal sections. **e** Azan staining of $Foxg1^{Cre}$; $Sox9^{flox/flox}$ trachea epithelium. **f** Representative 3D reconstructed images of epithelial-cell cluster (20 cells) of $Foxg1^{Cre}$; $Sox9^{flox/flox}$ or littermate control at E18.5. (left, apical view; middle, oblique view; right, lateral view). See also Supplementary Movie 7. **g** Apical-surface area at E18.5. center line, box limits, and whiskers represent mean, 25 and 75% confidence limits, and min and max values, respectively. Numbers represent means. (Control, n = 119, $Foxg1^{Cre}$; $Sox9^{flox/flox}$, n = 107). **h** Pseudostratification index at E18.5. center line, box limits, and whiskers represent mean, 25 and 75% confidence limits, and min and max values, respectively. Numbers represent means (Control, n = 130; $Foxg1^{Cre}$; $Sox9^{flox/flox}$, n = 84). (**i**) Azan staining of developing esophageal epithelium. P values (**P < 0.01, *P < 0.05) show the significance with Student's t test (**b**, **c**, **g**, **h**). NS not significant. Scale bar = 500 μm (**a**), 50 μm (**d**), 10 μm (**f**), 5 μm (**i**)

epithelial rearrangement in phase-2 tubulogenesis, while Sox9-positive chondrocytes regulate the epithelial-cell differentiation via secretary factors. These analyses suggest that epithelial-cell reshaping requires the structure of the ring cartilage and epithelial quiescence induced at the appropriate time, as revealed by $Nkx2.1^{CreERt2}$; $LSL\text{-}Kras^{G12D}$ (Fig. 2f). Immunohistochemical analyses failed to detect a clear reshaping of cell–cell adherence or of cytoskeletal or polarity-related proteins during phase 2 (Supplementary Fig. 15), indicating that cartilage ring development is most likely the major factor inducing the epithelial-cell reshaping. In developing esophagus that does not have cartilage, epithelial-cell reshaping was not detected during E14.5–18.5 (Fig. 6i). Thus, cartilage development distinguishes not only tube shape but also epithelial structure of the trachea.

To examine whether Wnt5a-Ror2 signaling plays a role in phase-2 tubulogenesis, we assessed the diameter, cell proliferation, apical enlargement, and apical emergence in $Wnt5a^{-/-}$ embryos at E18.5. Consistent with the phenotype at E12.5, the diameter of the $Wnt5a^{-/-}$ trachea was increased (Figs. 3a, 5e, Supplementary Fig. 16a), whereas the $Foxg1^{Cre}$, $Ror2^{flox/flox}$ trachea had a diameter similar to that of controls at E18.5 (Supplementary Fig. 16b). These data indicated that Ror2 may not be an exclusive receptor for Wnt5a in phase 2. We further examined the diameter in the Ror1/2 double knockout and observe no morphological difference between $Foxg1^{Cre}$, $Ror2^{flox/flox}$, and $Foxg1^{Cre}$, $Ror1/2^{flox/flox}$ at E18.5 (Supplementary Fig. 16b, c), suggesting that Ror1/2 are not involved in phase 2 tubulogenesis. Given that $Wnt5a^{-/-}$ showed a thickened diameter phenotype at E18.5, we next examined whether Wnt5a might regulate the luminal formation in phase 2 tubulogenesis (Supplementary Fig. 16d, e). Although the area of the apical surface of the epithelial cells in the $Wnt5a^{-/-}$ trachea was smaller than that in the control, apical enlargement occurred in the $Wnt5a^{-/-}$ trachea (Supplementary Fig. 16d). In contrast, the apical emergence was inhibited in the $Wnt5a^{-/-}$ trachea (Supplementary Fig. 16e), suggesting that Wnt5a is involved in apical rearrangement. Given that the $Wnt5a^{-/-}$ trachea showed elongation defects from E11.5 to E14.5, it is possible that the limited length of the $Wnt5a^{-/-}$ trachea restricts the epithelial reshaping.

## Discussion

In this study, we established that the SM progenitor polarization and cartilage differentiation differentially regulate the length and diameter of the murine trachea. This arises during tubulogenesis with Wnt5a-Ror2 signaling regulating proper SM morphogenesis by orchestrating the radial SM progenitor cell polarization in the subepithelium. This provides the circumferential restraint that directs tube elongation. This is followed by developing cartilage rings expanding the diameter and indirectly inducing epithelial-cell reshaping.

Although the mechanism by which Wnt5a-Ror2 signaling synchronizes radial SM progenitor polarization awaits further investigation, we expect that an additional epithelial factor is required to direct the polarity toward the epithelium and that the function of Wnt5a-Ror2 is permissive in SM progenitor cells. Wnt5a is reported to be critical for the development of tracheal cartilage[25]. In the present study, whole-body Wnt5a deletion did not result in cartilage ring malformation, while it decreased the number of cartilage rings. Thus, we think that the cartilage malformation of the $Wls^{flox/flox}$; $Dermo1^{Cre}$ embryos in the previous report resulted from a defect in canonical Wnt signaling.

Our finding demonstrates that Wnt5a-Ror2 signaling controls SM progenitor cell migration toward epithelium by synchronizing radial cell polarity. Disruption of the polarity resulted in altered

morphogenesis of SM progenitor cells at subepithelium. Given that intercalating movement is a potential mechanism by which developing tissue is elongated[35], intercalation of SM-cell at subepithelium might contribute to trachea tube elongation. We also propose the idea that circumferential cell–cell connection of these subepithelial SM progenitors limits radial expansion of developing trachea and promotes tube elongation. In the present study, we failed to provide direct evidence showing that the intercalation and/or mechanical force in SM elongates the developing trachea. On the other hand, we have also reported that the potassium channel KCNJ13 is a critical regulator of tracheal tube elongation[36]. KCNJ13 controls the polarity and alignment of tracheal SM cells by activating AKT signaling, which is essential for the activation of SM actin. These data also support the idea that circumferential restriction is involved in trachea tube elongation. Contractility-dependent trachea tubulogenesis is also observed in Drosophila[37]. In Drosophila tracheal tubes, actin filaments concentrate at the apical cortex of epithelial cells and form circumferential supracellular actin rings like those of mouse tracheal SM in the present study. The distribution pattern and contractility of the actin rings can influence tracheal tube morphology. In addition, radial cell intercalation is known to contribute to tissue elongation across species[35], implying that the developmental strategy behind mouse tracheal tubulogenesis might be evolutionarily conserved.

In general, layers of visceral SM cells line the peripheries of organs and tubes in the body. Although the pathological characteristics of visceral SM tissue has been investigated in human diseases, such as the SM remodeling in chronic inflammation, only a few reports have addressed the physiological significance of SM in organogenesis. In the branching morphogenesis of the bronchiole, the local differentiation of SM cells segregates nascent distal buds by creating epithelial clefts[38]. In gut development, the sequential differentiation of SM layers generates epithelial villi by generating compressive stresses that lead to epithelial buckling and folding in chick[11], whereas local proliferation plays a major role in mouse[12]. Here, we revealed an essential role of SM tissue in the directed tube elongation of developing trachea and esophagus along the anterior–posterior axis in mouse.

Epithelial-cell reshaping of developing endoderm is also reported in the intestine[39], in which densely populated pseudostratified columnar epithelium is rearranged to columnar epithelium during villus formation in the mid- to late-gastrulation stage. This process is regulated by cytoskeletal organization on epithelial cells. In contrast to the epithelial complexity of the intestinal lumen, the trachea develops a smooth epithelium, which supports efficient ventilation by reducing turbulence. In this study, we also failed to detect a change in the expression or distribution of cell–cell adherent complexes, cytoskeleton, or cell polarity-related molecules in the epithelial cells during cell reshaping (Supplementary Fig. 15). However, it is still possible that endogenous epithelial factors are involved in trachea tubulogenesis. These factors should be explored in future studies. Instead, we found that proliferation was reduced, and that the radial growth of cartilage expanded the luminal surface area, indicating that epithelial-cell reshaping is achieved by the timed induction of epithelial quiescence and cartilage development. While the ablation of tracheal chondrocytes impacts epithelial differentiation[27], our $Col2a1^{-/-}$ mice demonstrated that Sox9-positive chondrocytes are not sufficient to induce epithelial-cell reshaping. This observation indicates that the structure and rigidity of the cartilage, rather than paracrine signals from the chondrocytes, is required for epithelial reshaping. Our findings reveal a strategy for straight-organ tubulogenesis driven by SM progenitor polarization and ring cartilage development.

## Methods

**Animals**. The animals were housed in environmentally controlled rooms, and all the experimental procedures using animals were approved by the Institutional Animal Care and Use Committee of RIKEN Kobe Branch. We handled the mice in accordance with the ethics guidelines of the institute. To minimize tissue deformation, in all experiments, embryos were fixed in 4% paraformaldehyde/ phosphate buffered saline (PBS) overnight at 4 °C, and then tracheas were dissected. $SHH^{Cre}$, $Foxg1^{Cre}$, $Tagln^{Cre}$, $R26R^{RG}$, $R26R^{H2B-EGFP}$, $LSL-Kras^{G12D}$, $Sox9^{flox}$, $Wnt5a^{-/-}$, $Wnt5a^{flox}$, $Ror1^{-/-}$, $Ror2^{-/-}$, $Ror1^{flox}$, $Ror2^{flox}$, $Srf^{flox}$ mice have been generated previously[23,40–48]. $Nkx2.1^{CreERT2}$ mouse was provided by Jackson lab. For mosaic labeling of epithelial cells, $Nkx2.1^{CreERT2}$ mice were crossed with $R26R^{ACTB-mTmG/+}$ mice. Tamoxifen (0.5 mg; Sigma-Aldrich) was intraperitoneally injected into pregnant mice at E10.5. Embryos were collected at E15.5. To induce the epithelial-cell proliferation, $Nkx2.1^{CreERT2}$ mice were crossed with $R26R^{KrasLSLG12D/+}$ or $R26R^{KrasLSLG12D/mTmG}$ mice. Tamoxifen (5 mg) was intraperitoneally injected into pregnant mice three times at E14.5, 15.5, and 16.5. Embryos were collected from E14.5 to E18.5.

$Nkx2.1^{dTomato}$ mouse and $Col2a1^{-/-}$ mouse were established with CRISPR/ Cas9 genome editing technologies in zygotes. For $Nkx2.1^{dTomato}$ mouse, the frozen eggs were obtained by in vitro fertilization between C57BL/6 oocytes and heterozygous $Nkx2.1^{CreERt2}$ sperm. To insert a dTomato cassette with a FRT sequence in front of CreER allele at Nkx2.1-CreER allele, the mixture of 50 ng μl$^{-1}$ crRNA, 100 ng μl$^{-1}$ tracrRNA, 10 ng μl$^{-1}$ donor plasmid, 100 ng μl$^{-1}$ Cas9 protein, and 100 ng μl$^{-1}$ Cas9 mRNA was injected into the pronuclei. A total of 229 zygotes were injected, and 223 were transferred into pseudopregnant Institute of Cancer Research (ICR) female mice. Fifty-four pups were obtained and one out of them carried the designed $Nkx2.1^{dTomato-FRT-CreER-FRT}$ allele. The germline transmission was confirmed by crossing with C57BL/6, and then the CreER cassette was removed by mating the male mice with ACTB:Flpe B6 female mice[49].

For Col2a1 knockout mice, the frozen eggs were obtained by in vitro fertilization between C57BL/6 oocytes and heterozygous $Nkx2.1^{mTomato}$. Pronuclear stage frozen eggs were used in this study, and electroporation was performed[50]. For electroporation, the eggs were transferred into Opti-MEM I containing 25 ng μl$^{-1}$ crRNA1, crRNA2, 100 ng μl$^{-1}$ tracrRNA, 400 ng μl$^{-1}$ bridge oligo, and 100 ng μl$^{-1}$ Cas9 protein (TAKARA). CUY21 and LF501PT1-10 platinum plate electrode (BEX Co. Ltd.) were used, and the electroporation conditions were 30 V (3 ms ON + 97 ms OFF) 7 times. After electroporation, the zygotes were transferred into pseudopregnant ICR female mice, and then the embryos were collected at E18.5 stage. Genotype was determined by genetic PCR with combination of following primers; F1, 5′-GCAAAGCAGTATTCACCTTCC G-3′, R1: 5′- CTCGAACTCGTGGCCGTTCATG-3′, F2: 5′- GGGCTCTATGGC TTCTGAGGC-3′, R2: 5′- CACCTGCGGTGCTAACCAGCG-3′. Genotype was determined by genetic PCR with combination of following primers; Col2a1 F1, 5′- TGCCACTTAGTTTCACCAACTACC-3′ Col2a1 R1, 5′-GATGAGGGCTTCCAT ACATCCTTA-3′, Col2a1 F2, 5′-TTGCAGGAAAATTAGGGCCAAAG-3′ Col2a1 R2, 5′-GCAAAACATGGACCAGTGAAATGG-3′ (See Supplementary Fig. 13a, b).

Guide RNA sequences were designed by CRISPR design developed in Zhang laboratory (http://crispr.mit.edu). The crRNA and tracrRNA sequences used for genome editing were synthesized as follows (Fasmac). Nkx2.1-CreER crRNA: GGGGCCCCCCCTCGAGGTCGA, Col2a1 crRNA1; GAAUCGUGAACCAGCGA UAAguuuuagagcuaugcuguuuug, Col2a1 crRNA2; CGCUCAUACAUACAAUCGC Gguuuuagagcuaugcuguuuug. tracrRNA: AAACAGCAUAGCAAGUUAAAAUAA GGCUAGUCCGUUAUCAACUUGAAAAAGUGGCACCGAGUCGGUGCU.

### Cumulative BrdU-incorporation assay

To label almost all of proliferating cells, BrdU (0.1 mg ml$^{-1}$, Sigma-Aldrich, B5002) were cumulatively injected into pregnant mice four times every 2 h before sacrifice. To measure the proliferating ratio, BrdU$^+$ cells were manually counted between the 1st and 12th cartilage region, based on the sections immunostained for BrdU or pHH3. Before E13.5, when the cartilage did not appear yet, the cells were counted in the entire region of the trachea in sagittal sections.

### Measurement of epithelial-cell number in entire trachea

$SHH^{Cre}$; $R26R^{H2B-EGFP}$ or $SHH^{Cre}$; $R26R^{H2B-mCherry}$ tracheas were cleared by CUBIC reagent[51]. Fixed tracheas were soaked in CUBIC reagent-1 with gentle rocking at 37 °C for 2 days. After washing with PBS, samples were incubated in CUBIC-2 for 3 days. Images were acquired with LSM710 confocal laser scanning microscope with ×25/0.8 NA Imm objectives (Carl Zeiss). Images were acquired under the following conditions: 1024 × 1024 pixel size in x, y 0.415 μm, 2.400 μm z-section. Image processing was performed using a ZEN2012 (Carl Zeiss), and Imaris8 (Bitplane). A series of images (Z-stack) were reconstructed in Imaris. To distinguish individual nuclei, we newly created a spot layer and manually removed spots derived from non-specific signals. Finally, the total number of epithelial cells was automatically counted.

### Histology

For azan staining, tissue was fixed in Bouin's fixative solution (Sigma-Aldrich, HT10132). After dehydration, samples were embedded in paraffin and cut into 4 μm. Staining were performed according to manufacture's instruction (MUTO PURE CHEMICALS Co., Ltd.).

### Measurement of the developing trachea size

The length in Figs. 1c, 2h, 3b, g, j, 6b, Supplementary Fig. 3b, 5h, 14e were determined by the straight line connecting pharynx to primary branch on 2D images. The diameter in Figs. 1d, 2i, 5f, g, 6c, Supplementary Fig. 3c, 14f, a, b were determined by the mediolateral straight line in middle region of the trachea on 2D images. Mouse embryos were fixed in 4% paraformaldehyde (PFA) overnight and the tracheas were prepared. Tracheas were imaged by Leica M205 FA.

A combination method of micro-CT scanning and 3D reconstruction were performed to obtain more accurate data set reflecting 3D morphology. The length in Fig. 1g was determined by the curve connecting pharynx to primary branch along luminal surface on 3D images. The internal circumference in Fig. 1h was determined by the curve surrounding luminal surface on 3D images. For micro-CT scanning mouse skins were peeled off to infiltrate solution. Embryos were fixed in Bouin's fixative solution overnight, and washed in 70% EtOH with saturated Li$_2$CO$_3$ (Nacalai Tesque, 20619-42). ScanXmate-E090S (Comscantecno) was used for micro-CT scanning. Before scanning, samples were soaked in contrast agent, 1% phosphotungstic acid/70% ethanol, with gentle rocking at room temperature. The images were 3D reconstructed by Imaris8.1.2. software. The 3D image of trachea region was cropped, and inverted in order to extract luminal structure. And then, surface properties were created to reconstruct 3D structure of the lumen. Based on these reconstructions, longitudinal length, circumferential length and surface area were measured.

### In situ hybridization

For frozen sections, tracheas were incubated in 30% sucrose before embedding in OCT compound. In situ hybridization was performed on frozen sections (16 μm). Wnt5a cDNA (Genbank; NM001256224.1 nucleotids 95-402) was amplified by following primers; 5′-ATA GTC GAC ATG GCT TTG GCC ACG TTT TT-3′ and 5′-ATA GAA TTC ATT TGC ATC ACC CTG CCA AA-3′, and subcloned to pBluescriptII SK. DIG-labeled cRNA probe was synthesized with mMESSAGE mMACHINE kit (Ambion) and DIG RNA labeling mix (Roche Life Science). Sections were permeabilized in 0.1% Triton X-100/PBS for 30 min and blocked in acetylation buffer. After prehybridization, sections were hybridized with DIG-labeled RNA probe (500 ng mL$^{-1}$) overnight at 65 °C. Subsequently, sections were washed, and incubated with anti-DIG-AP antibodies (1:1000, Roche Life Science, 11 093 274 910), and colored with BM-purple (Roche Life Science, 11442074001).

### Immunostaining

For paraffin sections, tracheas were dehydrated and embedded in paraffin. Sections (6 or 8 μm) were stained with primary antibodies listed in Supplementary Table 1. Secondary antibody conjugated with Alexa Fluor 488/594/ 647 were used. F-actin was visualized by Alexa Fluor 488 Phalloidin (1:1000, Thermo Fisher Scientific, A12379). DAPI (Nacalai tesque) and TOPRO-3 (1:1000, Thermo Fisher Scientific, T3605) were used as nuclear counterstain. Detailed procedure were listed in Supplementary table 1.

Images were obtained with LSM710 or LSM780 confocal laser scanning microscope with ×63/1.4 NA Oil and ×25/0.8 NA Imm objectives (Carl Zeiss).

### Whole trachea staining

For epithelium staining, after fixation, samples were blocked in 5% normal donkey serum/PBS, and incubated with CDH1 antibody (1/800, Cell Signaling Technology). The following day, samples were washed five times for 1 h, and then, incubated overnight with secondary antibody conjugated with Alexa Fluor 488 (1/500, Thermo Fisher Scientific) and TOPRO-3 (1/500, Thermo Fisher Scientific). After washing, tracheas were cut in the coronal plane, and their ventral epithelium was imaged. For SM staining, samples were blocked in 5% fetal bovine serum (FBS) (v/v), 3% bovine serum albumin (BSA) (w/v), 0.5% Triton X-100/PBS (v/v), and incubated with Cy3 conjugated SMA antibody (1/400, Sigma-Aldrich) and TOPRO-3 (1/500, Thermo Fisher Scientific). The following day, samples were washed five times for 1 h, and then fixed in 4% PFA/PBS for 1 h again. Samples were soaked into CUBIC reagent-1 with gentle rocking at 37 °C overnight. After washing with PBS, samples were incubated with CUBIC reagent-2 at 37 °C overnight. For cartilage staining, samples were blocked in 5% FBS (v/v), 3% BSA (w/v), 0.5% Triton X-100/PBS (v/v), and incubated with Sox9 antibody (1/400, Millipore) and TOPRO-3 (1/500, Thermo Fisher Scientific). The following day, samples were incubated with secondary antibody conjugated with Alexa Fluor 488 (1:500) at 4 °C overnight, and washed five times for 1 h, and then fixed in 4% PFA/PBS for 1 h again. Samples were soaked into CUBIC reagent-1 with gentle rocking at 37 °C overnight. After washing with PBS, samples were incubated with CUBIC reagent-2 at 37 °C overnight. Cleared tracheas were imaged by confocal LSM710 confocal laser scanning microscope with ×25/0.8 NA Imm objectives (Carl Zeiss).

### 3D culture of epithelial cells and SM cells from the trachea

The epithelial and SM cells were prepared from $Tagln^{Cre}$; $R26R^{ACTB-mTmG/+}$ or $Tagln^{Cre}$; $R26R^{ACTB-mTmG/+}$; $Wnt5a^{-/-}$ mouse tracheas at E12.5, as follows. The tracheas were digested in 0.25% trypsin for 10 min. The cells were passed through a 40-μm cell strainer, and then trapped in 50% Matrigel/mTEC-plus on a 35-mm glass-bottom dish. The cells embedded in Matrigel were covered with mTEC-plus medium supplemented with 10 μm Y-27632. After day 2, the cells were maintained in mTEC-plus without Y-27632.

For time-lapse imaging, the cells on day 3 were imaged every 1 h for 24 h by an Olympus LCV-MPE (multiphoton/incubator) or an Olympus LCV-CSUW1. The images were acquired with the following conditions: $1024 \times 1024$ pixel size in x, y 0.415 μm, 2.0 μm z-section. Image processing was performed using Image J (NIH), and Imaris 8.1.2 (Bitplane).

For quantification, cells were manually traced by Image J. The approaching distance was defined as the remainder after subtracting the final distance between the SM cell and the epithelial-cell sphere from the initial distance (Supplementary Fig. 8). Directionality was indicated by the angle between the two vectors $\overline{AB}$ and $\overline{AA'}$ (Supplementary Fig. 8), where A was the initial position of the SM cell, B was the initial position of the sphere, and A' was the final position of the SM cell.

**Peristalsis analysis.** After dissection of the internal organs including the trachea from $Wnt5a^{-/-}$ and control littermates at E12.5, the organs were transferred onto the membrane (Nucleopore Track-Etch Membrane (8 μm pore size), Whatman) in DMEM/Ham's F12 medium with 5% FBS. For live imaging, the trachea in the organs was cultured for 24 h, and then imaged for 10 min every 1 s by DeltaVision2 (Olympus). For measurement of the max contractility was measured every second during contraction. We defined maximum contractility as the rate of minimum diameter to maximum diameter.

**Quantification of epithelial rearrangement.** $SHH^{Cre}$; $R26R^{RG}$ and $SHH^{Cre}$; $R26R^{RG}$; $Wnt5a^{-/-}$ tracheas were fixed in 4% PFA/PBS and coronally opened in PBS(−). For the analyses of $Nkx2.1^{mTomato}$; $Col2a1^{-/-}$ mice, mice were fixed in 4% PFA at 4 °C overnight and tracheas were collected. After wash with PBS(−), tracheas were made transparent by sequential incubation in CUBIC regent 1/2. Epithelium of the ventral side was imaged by a confocal LSM710 confocal laser scanning microscope with ×25/0.8 NA Imm objectives (Carl Zeiss). Images were acquired under the following conditions: $1024 \times 1024$ pixel size in x, y 0.332 μm, 2.000 μm z-section. Images were processed using ZEN (Carl Zeiss), Imaris (Bitplane), and Image J (NIH). For the analyses of $Foxg1^{Cre}$; $Sox9^{flox/flox}$ mice, tracheas were stained for CDH1 and TOPRO-3 as described above. For apical area measurement, we selected a relatively flat region to avoid the influence by tissue folds and manually followed the apical shape contour. For the luminal cell ratio, we manually counted the numbers of luminal cells, which were attached to the lumen, and the non-apical basal cells. The psuedostratification index was determined as described previously with some modification[33]. In optical images along the AP-axis, the contours of the nucleus were followed to calculate the x, y-coordinates of the centroid. The distance between the nucleic centroid and the basal membrane was measured as the pseudostratification index.

**Quantification of SM-cell polarity.** All quantification was based on the sagittal sections stained for GM130, Ror2, and TOPRO-3 as described above. The contours of the plasma membrane in Ror2-expressing SM cells were manually traced to calculate the x, y-coordinates of the centroid in sagittal sections. The orientation of SM cells was judged by the relative x, y-coordinates of the nucleic centroid and Golgi apparatus. The distribution of the orientations was expressed as rose charts.

**Statistics.** Statistical analyses were performed with Excel2013 (Microsoft). For multiple comparisons, statistic significance was determined by the Tukey's method. For paired comparisons, statistic significance was determined by Student's or Welch's two-tailed $t$ test for equal or unequal variance populations, respectively.

**Data availability.** The authors declare that all data supporting the findings of this study are available within the article and its Supplementary Information files or from the corresponding author upon reasonable request.

The datasets generated during the current studies are available in the System Science of Biological Dynamics (SSBD) database (http://ssbd.qbic.riken.jp/).

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

## Acknowledgments

We thank Terry Yamaguchi and Reiko Ajima for the Wnt5a conditional flox mice, Michael Eldon Greenberg for Ror1 conditional flox mice, Hiroki Ueda for the Sox9 conditional flox mice, Mark Krasnow and Maya Kumar for the Tbx4-Cre mice, Steven L. Brody for the anti-Foxj1 antibody, and the CDB 4D Imaging Unit and Animal Resource Development Unit as well as Cellular Dynamics Analysis for support. We also thank Masako Yamaguchi for supporting IMARIS works; Masatoshi Takeichi, Shigeo Hayashi, Didier Stainier and Brigid Hogan for primary reading; and Chisa Matsuoka and Yuka Noda for general technical support. These studies were supported by funding from Grants-in-Aid for Young Scientists (B) (No. 17K15133) of the Ministry of Education, Culture, Sports, Science and Technology, Japan (K.K.), and The Takeda Science Foundation for the Life Science (M.M.), and RIKEN Single Cell Project (M.M.) Scientific Research on Innovative Areas of the Ministry of Education, Culture, Sports, Science and Technology, Japan (No. 23112004) (M.M.).

## Author contributions

K.K. and M.M. designed the project and performed experiments with the aid of A.Y., M.T. performed the micro-CT experiments. M.N. and Y.M. provided Ror2 antibody and Ror1, 2 conditional KO mice, and supported data interpretation. Col2a1 KO mice and Nkx2.1-mTomato were generated by T.A. and M.S., K.K. and M.M. wrote the manuscript with the contribution of all the authors.

## Additional information

**Competing interest** The authors declare no competing interests.

