## [Peer Review File · Nature Communications]

Reviewers' comments:

Reviewer #1 (Remarks to the Author):

Here the authors investigate tube morphogenesis focusing on the trachea. Morphometric analysis of trachea development and tracking of cellular dynamics at the single cell level provide a two phase model for tube development with the first phase dominated by tube elongation that is driven by Wnt5a-Ror signaling in the surrounding mesoderm that serves to drive radially polarized organization of smooth muscle cells that in turn restricts elongation to the AP axis. The second phase requires cartilage development that expands the tube diameter and drives epithelial cell reshaping. Overall, this is a very well conceived and executed manuscript addressing a poorly understood area of biology that reveals a novel mechanism to drive tube elongation. The findings will have important implications in understanding tube morphogenesis in other organs. I have some comments that I think would be important to address:

1) In Figure 1 the authors take morphometric measurements of tube development and measure tube length, circumference and lumen surface area, as well as proliferation. The measurements are carefully made, but the analysis of fold changes between time points is not the right way to analyze these data if they want to make conclusions about the rate of change. For example they discuss the changing surface area (Fig. 1i) and conclude that between 14.5 and 16.5 surface area expansion slows down, but then accelerates between 16.5 and 18.5. Actually conclusions about rate of change of surface area expansion must be made based on the slope of the curve, not the fold change between the points. Thus, looking at the top panel it is clear that the rate of surface area expansion is relatively constant during phase 1 between 12.5-16.5 (ie-there is a straight line) but then accelerates somewhere between 16.5 and 18.5, such that the rate of surface area expansion is increased at the late phase.

The same problem crops up in panel I, where the rate of change of epithelial numbers stays relatively constant throughout the time course, which is consistent with the reduced proliferation rate later in development.

This analysis is much more compelling in my opinion and the accelerated lumen area nicely correlates with the diameter expansion phase and is supported by the nice cell shape analysis performed in Figure 2.

2) Based on the alternative analysis above, it would be interesting to get a more careful assessment of surface area expansion at 15.5 and 17.5 to better map the transition point.

3) In Figure 2 they induce proliferation with activated K-ras. The conclusions that the epithelial layer was defective but the tube was intact is not obvious from the presented data. Minimally the authors need to count cells (and ideally proliferation rates) to confirm that epithelial numbers were increased by K-ras expression. Ideally morphometric analyses (as in Figure 1) and cell shape analyses (as in Figure 2a) need to be performed to provide the data backing up the authors' claims.

4) The genetic analysis demonstrating that Wnt5a from the mesoderm is critical for tube elongation is very elegant, but again, I'd like to see analysis of epithelial cell numbers and proliferation rates, which, given the results and conclusions of Figure 1, should be reduced upon Wnt5a loss.

5) The Ror analysis is very nice and convincingly shows the interesting result that Wnt5a signaling in mesenchyme underlying the epithelia is what dictates the phase 1 elongation of the trachea. Quantitation of trachea lengths in Fig. 3i is important to include.

6) The final studies on cartilage requirement to expand the lumen and the apical surface seem self

evident in this model so it is of interest to show it. The cartilage deficient tracheas show a convoluted surface. Is the total surface area different compared to in the presence of cartilage?

Reviewer #2 (Remarks to the Author):

This manuscript describes a quantitative analysis of tracheal morphogenesis in mice and presents evidence for a new model of how the fetal trachea grows. In general the work provides novel insight in to an important understudied aspect of respiratory system development, which when defective can result in severe birth defects. Strengths of the study include the time course quantification of epithelial proliferation, apical expansion and apical emergence, which are rigorous and novel.

In addition the observation that conditional deletion of Wnt5a and Ror2 reveal specific function in trachea elongation will be of interest to the field. Despite these advances a major limitation in the study is that the current experiments and data do not really support the authors mechanistic conclusions that biomechanical forces from the mesenchyme control epithelial behavior. They have good correlative data but they have not really demonstrated a causative mechanism. The authors either have to modify the conclusions, which would reduce the impact significantly, of the study or provide more direct evidence testing their model.

Major concerns:

The authors claim that biomechanical forces controlled first by the dorsal smooth muscle and then by the tracheal cartilage control epithelial behavior to shape the trachea – but this has not really been demonstrated.

1. The observations that: 1) the trachea is shorter in Wnt5a-Ror2 mutants, 2) the dorsal SM is polarized in a Wnt5-Ror2 dependent manner and 3) that the fetal trachea exhibits peristalsis are all well documented. From these correlations the authors claimed that “radial polarization of smooth muscle cells”.. “regulates tube elongation”. However there is not direct evidence to support this causative relationship. It is equally possible that Wnt5a-Ror2 expressing mesenchyme signals to the epithelium to control proliferation and apical expansion, and that mesenchymal polarization and peristalsis have nothing to do with the epithelial morphogenesis. The authors must quantify the epithelial proliferation, apical expansion and apical emergence when Wnt5a-Ror2 is specifically deleted in the SM. The authors should also directly disrupt the SM band and/or peristalsis and show that this causes shortening, perhaps by cutting the SM in culture or even better by genetically or chemical disrupting SM polarization and/or contractility and show that this impacts epithelial behavior and tracheal length. Without these data the mechanistic conclusions are not founded.

2. Similarly the authors conclude that “cartilage development helps expand the tube diameter that drives epithelial cell reshaping”, but they have no direct evidence to support this. While the epithelial morphogenesis is disrupted in the Foxg1:cre; Sox9f/f embryos and these fail to form cartilage exhibiting a collapsed trachea, this data does not demonstrate that it is the biomechanical force of the cartilage that drives the epithelial behavior. It is equally possible that the Sox9 expressing mesenchyme signals to the epithelium to regulate apical expansion. In order to test their model the authors must directly disrupt cartilage stiffness, perhaps by cutting the cartilage in cultures models or by a Dermo1:Cre; Col2a1f/f model.

3. Foxg1;Cre is well known to recombine in both the foregut epithelium and mesenchyme, thus the author contention that this is a mesenchyme specific deletion is unfounded. Since epithelial Wnt5a has been shown to be critical for tracheal cartilage (PMID: 25727890) this should be directly addressed. The Dermo1;Cre might be better.

Minor problems:

The observation that cartilage promotes of epithelial maturation (albeit from a different viewpoint) is already published (PMID 27941074) and should be addressed.

The phrase proliferation-independent is misleading since there is a low level of proliferation, which is probably important.

Fig2. Kras overexpression experiment needs further validation that excess proliferation is indeed achieved, and that does not affect length nor diameter.

Reviewer #3 (Remarks to the Author):

This manuscript used multiple mouse models to investigate the mechanisms controlling the tube structure of the trachea. By disrupting Wnt5a-Ror2 signaling specifically in the mesenchyme and epithelium, the authors found that it is the radical polarization of the mesenchyme that contributes to the elongation of the trachea. They also conclude that epithelial proliferation is not required for tubulogenesis by examining NKX2.1CreERT2; LSL-KrasG12D mice that present excessive proliferation of the trachea. Furthermore, the authors discovered that the cartilage rings surrounding the trachea play an essential role in the tube diameter expansion but not elongation. Overall, this manuscript carefully describes phenotypic changes in response to genetic manipulations and provides critical insights into the mechanism underlying the tracheal tubulogenesis. The results are informative and shed new lights into the understanding of foregut organogenesis. These findings also provide insights into developmental disorders that occur to the upper airways. That said, several concerns need to be addressed.

1. The authors showed that smooth muscle (SM) cell polarization was changed in Wnt5a/Ror2 mutants and suggested that the cell polarization contributes to tube elongation. Was the proliferation of SM cells affected by disrupting Wnt5a-Ror2 signaling?
2. What is the tracheal phenotype of Foxg1Cre/+ and Foxg1 cre/cre embryos? This information needs to be provided.
3. The manuscript stated that epithelial cell reshaping requires mechanical expansion force provided by mesenchyme. Besides mechanical force, is any molecular contribution from the tracheal cartilage? The authors need to provide some discussion at least.

Reviewer #4 (Remarks to the Author):

In this manuscript the authors show that different mechanisms regulating the length and diameter of the murine trachea. First, they find that trachea development progresses in two different phases. In a first phase, mesenchymal Wnt5a-Ror2 signalling controls synchronized radial polarization of smooth muscle (SM) cells to control tube elongation. This radial polarization targets SM cell toward the epithelium, and the resulting subepithelial SM morphogenesis limits tube elongation to the antero-posterior axis. This initial phase is highly proliferative, both in the stromal and the epithelial compartment. Subsequently, cartilage development helps expand the tube diameter that drives epithelial cell reshaping to generate the optimal lumen shape. Their data seem to fit in a model in which Wnt-mediated SM polarized migration and ring cartilage

development drive organ tubulogenesis.

Overall, the findings provide by the authors in this study are certainly interesting, novel and of high quality, and thus, could be potentially interesting for broad readers of Nature Communications. However, the mechanistic insights are very limited. It is already known that wnt-ror signalling controls the elongation of the mouse intestine by a mechanism of cell intercalation, which is regulated by PCP, information that is not included by the authors. It is important to acknowledge as a possibility that the mechanism that the authors have described in the elongation phase of the tracheal tubulogenesis (wnt5a-ror2 dependent SM migration) might be analogous to gut elongation, which is a PCP-driven cell intercalation mechanism. In addition the authors should address some important major and minor points before publication.

RESULTS

1 Two step tracheal tubulogenesis:

- o Change fig 1 title (three-step to two-step)

- o Please explain the difference between length quantifications from E16.5 to E18.5 described in Fig 1c and 1g

2-Non-proliferative luminal enlargement:

- o Phase 2 is also proliferative, although in a reduced manner (x1.31). The authors should change the title to "low-proliferative" or eliminate any mention to proliferation on it.

- o The authors should demonstrate that proliferation increase after induction with tamoxifen in Nkx2.1CreERT2; LSL-KrasG12D mice (Fig 2f), and quantify this proliferation.

3-Wnt5a-Ror2 signalling extends the tube:

- o The authors already show Wnt5 and Ror2 is expressed only in the stromal compartment (figures 1c and 1h). It was interesting to confirm that conditional deletion of Wnt5a from epithelium gave a similar phenotype that controls (Fig 3d), but the conditional deletion of Ror2 from epithelium (Fig 3i) is not relevant, and does not provide any crucial information. I would suggest relocating to supplemental or eliminating.

- o The authors should better characterize the tube diameter increase (phase 2 of tubulogenesis) in Wnt5a^{-/-} and Foxg1Cre; Ror2^{flox/flox} mice: tube diameter, proliferation, apical enlargement and emergence...

- o Minor point: change E14.5 to E11.5 when talking about Fig 3c in the text

4-Radial polarization of subepithelial cells arrange SM morphology:

- o Authors do not consider defects in cell intercalation as potential mechanism controlling tracheal morphogenesis despite their results support this hypothesis, which would also be backed by evidence in bibliography of the regulation by Wnt5a/Ror2 of the elongation of another tubular organ, the intestine, in mice during development through PCP (Cervantes et al., 2009, Yamada et al., 2010)

- o Fig 5a should be properly quantified

- o According to Fig 5g, Ror1/Ror2 KO tracheas show a significant difference in external diameter.

This result shows that Ror1 might indeed have an interesting role in the control of tracheal morphogenesis (which is in contradiction with authors statements about Fig 3e-g results) and that both receptor functions seem to be redundant. Authors should analyse tubulogenesis phase 2 in these mice.

5-Cartilage growth expands tube diameter:

- o The authors should also analyse the SM cells, not only epithelial cells

DISCUSSION

6- The statement "in this study, we did not detect any changes in expression or distribution of cell-cell adherent complexes, cytoskeleton or cell polarity-related molecules in epithelial cell during cell reshaping" is too bold and further experiments would be necessary to address this issue. A confocal image (Fig S7) with different markers is not sufficient to demonstrate this point, which is clearly relevant.

Reviewers' comments:

Reviewer #1 (Remarks to the Author):

Here the authors investigate tube morphogenesis focusing on the trachea. Morphometric analysis of trachea development and tracking of cellular dynamics at the single cell level provide a two phase model for tube development with the first phase dominated by tube elongation that is driven by Wnt5a-Ror signaling in the surrounding mesoderm that serves to drive radially polarized organization of smooth muscle cells that in turn restricts elongation to the AP axis. The second phase requires cartilage development that expands the tube diameter and drives epithelial cell reshaping. Overall, this is a very well conceived and executed manuscript addressing a poorly understood area of biology that reveals a novel mechanism to drive tube elongation. The findings will have important implications in understanding tube morphogenesis in other organs. I have some comments that I think would be important to address:

1) In Figure 1 the authors take morphometric measurements of tube development and measure tube length, circumference and lumen surface area, as well as proliferation. The measurements are carefully made, but the analysis of fold changes between time points is not the right way to analyze these data if they want to make conclusions about the rate of change. For example they discuss the changing surface area (Fig. 1i) and conclude that between 14.5 and 16.5 surface area expansion slows down, but then accelerates between 16.5 and 18.5. Actually conclusions about rate of change of surface area expansion must be made based on the slope of the curve, not the fold change between the points. Thus, looking at the top panel it is clear that the rate of surface area expansion is relatively constant during phase 1 between 12.5-16.5 (ie-there is a straight line) but then accelerates somewhere between 16.5 and 18.5, such that the rate of surface area expansion is increased at the late phase.

The same problem crops up in panel l, where the rate of change of epithelial numbers stays relatively constant throughout the time course, which is consistent with the reduced proliferation rate later in development. This analysis is much more compelling in my opinion and the accelerated lumen area nicely correlates with the diameter expansion phase and is supported by the nice cell shape analysis performed in Figure 2.

Thank you for the comments. We agree that several methods could be used in this analysis for the better understanding. We calculated the slope for Fig. 1i and l, and then compared, see below. As the reviewer suggested, the slope showed similar increases between 12.5-14.5 and 14.5-16.5. However, we think this result does not reflect the biological character of epithelial cells because the 5,000 cells providing 25,000 cells during E12.5-14.5 have a different character from the 25,000 cells generating 45,000 cells during E14.5-16.5. Although epithelium provides 15,000 to 20,000 cells in both phases, all of cells divide more than two times in E12.5 to 14.5 while less than one-time cell division is enough during E14.5-16.5. In Fig 1i and l, we showed that surface area expansion is not exclusively supported by proliferation. We therefore believe that a fold change analysis would be better than a slop analysis to estimate the biological feature of epithelium in tube expansion.

2) Based on the alternative analysis above, it would be interesting to get a more careful assessment of surface area expansion at 15.5 and 17.5 to better map the transition point.

Using slope analysis above, the surface areas of E15.5 and 17.5 were estimated as below. This data, I'm afraid but, did not change our impression from current Fig. 1i. Thus, we show this data only in this revise letter.

3) In Figure 2 they induce proliferation with activated K-ras. The conclusions that the epithelial layer was defective but the tube was intact is not obvious from the presented data. Minimally the authors need to count cells (and ideally proliferation rates) to confirm that epithelial numbers were increased by K-ras expression. Ideally morphometric analyses (as in Figure 1) and cell shape analyses (as in Figure 2a) need to be performed to provide the data backing up the authors' claims.

We agree. Morphometric analyses of *Nkx2.1^{CreERT2}; LSL-KrasG12D* trachea were performed in this revision. We revealed that the Kras-trachea increase the cell number and proliferation rate of epithelium whereas the tube shapes were indistinguishable even after tamoxifen injection, during E16.5 to E18.5. We added these data to Supplementary Fig. 3 and mentioned in the manuscript.

4) The genetic analysis demonstrating that Wnt5a from the mesoderm is critical for tube elongation is very elegant, but again, I'd like to see analysis of epithelial cell numbers and proliferation rates, which, given the results and conclusions of Figure 1, should be reduced upon Wnt5a loss.

As suggested, we examined the cell number and proliferation rate of *Wnt5a^{-/-}* trachea. The *Wnt5a^{-/-}* showed the similar proliferation rate. In consistent with proliferating rate, the cell number per area was not significantly

changed in the mutant. We added these data to Supplementary Fig. 4a-c and mentioned in the manuscript.

5) The Ror analysis is very nice and convincingly shows the interesting result that Wnt5a signaling in mesenchyme underlying the epithelia is what dictates the phase 1 elongation of the trachea. Quantitation of trachea lengths in Fig. 3i is important to include.

We agree. We quantified the tube length of tracheas of *Foxg1^{Cre}*; and *SHH^{Cre}*; *Ror2^{flox/flox}* as well as *Foxg1^{Cre}*; *Wnt5a^{flox/flox}* to clarify a significant difference in the *Foxg1^{Cre}* mediated mesenchymal mutants. We added these data to Fig. 3j, Supplementary Fig. 5h.

6) The final studies on cartilage requirement to expand the lumen and the apical surface seem self evident in this model so it is of interest to show it. The cartilage deficient tracheas show a convoluted surface. Is the total surface area different compared to in the presence of cartilage?

The total surface area of no-cartilage mutant was measured by a combination method of micro-CT and reconstruction. The mutant showed 0.497-times smaller luminal area. We mentioned it in the manuscript.

Reviewer #2 (Remarks to the Author):

This manuscript describes a quantitative analysis of tracheal morphogenesis in mice and presents evidence for a new model of how the fetal trachea grows. In general the work provides novel insight in to an important understudied aspect of respiratory system development, which when defective can result in severe birth defects. Strengths of the study include the time course quantification of epithelial proliferation, apical expansion and apical emergence, which are rigorous and novel.

In addition the observation that conditional deletion of Wnt5a and Ror2 reveal specific function in trachea elongation will be of interest to the field. Despite these advances a major limitation in the study is that the current experiments and data do not really support the authors mechanistic conclusions that biomechanical forces from the mesenchyme control epithelial behavior. They have good correlative data but they have not really demonstrated a causative mechanism. The authors either have to modify the conclusions, which would reduce the impact significantly, of the study or provide more direct evidence testing their model.

Major concerns:

The authors claim that biomechanical forces controlled first by the dorsal smooth muscle and then by the tracheal cartilage control epithelial behavior to shape the trachea – but this has not really been demonstrated.

1. The observations that: 1) the trachea is shorter in Wnt5a-Ror2 mutants, 2) the dorsal SM is polarized in a Wnt5-Ror2 dependent manner and 3) that the fetal trachea exhibits peristalsis are all well documented. From these correlations the authors claimed that “radial polarization of smooth muscle cells”.. “regulates tube elongation”. However there is not direct evidence to support this causative relationship. It is equally possible that Wnt5a-Ror2 expressing mesenchyme signals to the epithelium to control proliferation and apical expansion, and that mesenchymal polarization and peristalsis have nothing to do with the epithelial morphogenesis. The authors must quantify the epithelial proliferation, apical expansion and apical emergence when Wnt5a-Ror2 is specifically deleted in the SM. The authors should also directly disrupt the SM band and/or peristalsis and show that this causes shortening, perhaps by cutting the SM in culture or even better by genetically or chemical disrupting SM polarization and/or contractility and show that this impacts epithelial behavior and tracheal length. Without these data the mechanistic conclusions are not founded.

We agree this reviewer. We added the quantitative data for the epithelial proliferation, apical expansion and apical emergence of Wnt5a KO into Supplementary Fig. 4 and 16.

We also agree that we do not provide any direct evidence suggesting that biomechanical forces controlled by the smooth muscle elongate tracheal tube. We moderated all our description claiming mechanical-force driven tube elongation. We also clearly mentioned in discussion that we failed to provide direct evidence showing mechanical force by smooth muscle elongates developing trachea in the present study. As for *ex vivo* experiment, because *ex vivo* culture method of embryonic trachea is not established yet, although we tried many times by ourselves, it is impossible to manipulate SM contractility in *in vitro*. Instead, we newly generated *Foxg1^{Cre}*, *Srf^{flox/flox}* mice to inhibit SM cell differentiation in developing trachea *in vivo*. While this mutant does not develop mature SM, the subepithelial radial cell polarization was intact. Unlike Wnt5a KO, this SM-null trachea showed normal tube elongation and proper subepithelial SM progenitor morphology. These observations suggested that acquisition of the radial cell polarity and/or proper SM progenitor morphogenesis are crucial for tracheal elongation. We added these data into Supplementary Fig. 9. We discussed about two possible mechanisms of tracheal tube elongation. One is SM progenitor migration directed by the radial cell polarity causes a convergent extension of subepithelium elongating tracheal tube. Another possibility is circumferential cell-cell connection of SM progenitor cells promotes anteroposterior tube elongation by limiting diameter expansion.

2. Similarly the authors conclude that “cartilage development helps expand the tube diameter that drives epithelial cell reshaping”, but they have no direct evidence to support this. While the epithelial morphogenesis is disrupted in the *Foxg1:cre;Sox9^{f/f}* embryos and these fail to form cartilage exhibiting a collapsed trachea, this data does not demonstrate that it is the biomechanical force of the cartilage that drives the epithelial behavior. It is equally possible that the Sox9 expressing mesenchyme signals to the epithelium to regulate apical expansion. In order to test their model the authors must directly disrupt cartilage stiffness, perhaps by cutting the cartilage in cultures models or by a *Dermo1:Cre;Col2a1^{f/f}* model.

Following this suggestion, we generated *Col2a1^{-/-}* mice using CRISPR-Cas9 genome editing technology. The *Col2a1^{-/-}*

trachea exhibited the narrowed trachea phenotype and failed epithelial reshaping as seen in *Foxg1^{Cre}; Sox9^{flox/flox}*. Thus, we again conclude that cartilage development helps expand the tube diameter that drives epithelial cell reshaping. We added these data into Supplementary Fig. 14.

3. *Foxg1;Cre* is well known to recombine in both the foregut epithelium and mesenchyme, thus the author contention that this is a mesenchyme specific deletion is unfounded. Since epithelial *Wnt5a* has been shown to be critical for tracheal cartilage (PMID: 25727890) this should be directly addressed. The *Dermo1;Cre* might be better.

We apologize that we did not mention that *Foxg1^{Cre}* can target a part of foregut epithelium. We revised the text and mentioned this point. To directly deny the possibility that epithelial recombination *Wnt5a* or *Ror2* flox alleles cause trachea malformation, we have conducted epithelial deletion for these genes and detected no obvious morphological defect. We further targeted *Wnt5a* in mesenchyme by using *Dermo1^{Cre}* as reviewer recommended. These data were shown in Fig. 3d and Supplementary Fig. 5g, h, k).

In our mouse colony, *Wnt5a* deletion did not result in the cartilage ring malformation while decreased the number of the rings (Supplementary Fig. 4d). Thus we think the cartilage malformation of *Wls^{flox/flox}; Dermo1^{Cre}* embryo in PMID: 25727890 is the defect of canonical Wnt signaling. We discussed this point in revised manuscript.

Minor problems:

The observation that cartilage promotes of epithelial maturation (albeit from a different viewpoint) is already published (PMID 27941074) and should be addressed.

Sorry for omitting this publication. We added a discussion about contribution of Sox9-positive chondrocyte.

“While ablation of tracheal chondrocyte impacts epithelial differentiation (Turcatel et al., 2017), our *Col2a1^{-/-}* demonstrated that Sox9-positive chondrocyte is not sufficient to induce epithelial cell reshaping. This observation indicates that the structure and rigidity of the cartilage, rather than paracrine from the chondrocytes, is required for epithelial reshaping.”

The phrase proliferation-independent is misleading since there is a low level of proliferation, which is probably important.

We apologize our misleading word choice. We revised.

Fig2. *Kras* overexpression experiment needs further validation that excess proliferation is indeed achieved, and that does not affect length nor diameter.

As we showed to Reviewer 1, we added these data into Supplementary Fig. 3.

Reviewer #3 (Remarks to the Author):

This manuscript used multiple mouse models to investigate the mechanisms controlling the tube structure of the trachea. By disrupting *Wnt5a-Ror2* signaling specifically in the mesenchyme and epithelium, the authors found that it is the radical polarization of the mesenchyme that contributes to the elongation of the trachea. They also conclude that epithelial proliferation is not required for tubulogenesis by examining *NKX2.1CreERT2; LSL-KrasG12D* mice that present excessive proliferation of the trachea. Furthermore, the authors discovered that the cartilage rings surrounding the trachea play an essential role in the tube diameter expansion but not elongation. Overall, this manuscript carefully describes phenotypic changes in response to genetic manipulations and provides critical insights into the mechanism underlying the tracheal tubulogenesis. The results are informative and shed new lights into the understanding of foregut organogenesis. These findings also provide insights into developmental disorders that occur to the upper airways. That said, several concerns need to be addressed.

1. The authors showed that smooth muscle (SM) cell polarization was changed in *Wnt5a/Ror2* mutants and suggested that the cell polarization contributes to tube elongation. Was the proliferation of SM cells affected by disrupting *Wnt5a-Ror2* signaling?

Thank you for the comment. We checked Ki67 expression in *Wnt5a/Ror2* mutants and detected no significant difference. We added these data into Supplementary Fig. 6 and mentioned in the text.

2. what is the tracheal phenotype of *Foxg1Cre/+* and *Foxg1 cre/cre* embryos? This information needs to be provided.

We examined *Foxg1^{Cre}* hetero and homo embryo and found no obvious morphological phenotype. We added these data into Supplementary Fig. 5c-f and mentioned in the text.

3. The manuscript stated that epithelial cell reshaping requires mechanical expansion force provided by mesenchyme. Besides mechanical force, is any molecular contribution from the tracheal cartilage? The authors need to provide some discussion at least.

Thank you for the suggestion. As we showed to Reviewer 2, we generated *Col2a1^{-/-}* embryo in which the cartilage development is disrupted while Sox9-positive chondrocytes are present, see Supplementary Fig. 14. The mutant

failed epithelial cell reshaping suggesting that a signal from the chondrocytes to epithelium is not sufficient to induce epithelial cell reshaping. So, the structure of ring cartilages could be required to achieve epithelial cell reshaping.

Reviewer #4 (Remarks to the Author):

In this manuscript the authors show that different mechanisms regulating the length and diameter of the murine trachea. First, they find that trachea development progresses in two different phases. In a first phase, mesenchymal Wnt5a–Ror2 signalling controls synchronized radial polarization of smooth muscle (SM) cells to control tube elongation. This radial polarization targets SM cell toward the epithelium, and the resulting subepithelial SM morphogenesis limits tube elongation to the antero-posterior axis. This initial phase is highly proliferative, both in the stromal and the epithelial compartment. Subsequently, cartilage development helps expand the tube diameter that drives epithelial cell reshaping to generate the optimal lumen shape. Their data seem to fit in a model in which Wnt-mediated SM polarized migration and ring cartilage development drive organ tubulogenesis.

Overall, the findings provide by the authors in this study are certainly interesting, novel and of high quality, and thus, could be potentially interesting for broad readers of Nature Communications. However, the mechanistic insights are very limited. It is already known that wnt-ror signalling controls the elongation of the mouse intestine by a mechanism of cell intercalation, which is regulated by PCP, information that is not included by the authors. It is important to acknowledge as a possibility that the mechanism that the authors have described in the elongation phase of the tracheal tubulogenesis (wnt5a-ror2 dependent SM migration) might be analogous to gut elongation, which is a PCP-driven cell intercalation mechanism. In addition the authors should address some important major and minor points before publication.

RESULTS

1 Two step tracheal tubulogenesis:

o Change fig 1 title (three-step to two-step)

Thank you. We revised.

o Please explain the difference between length quantifications from E16.5 to E18.5 described in Fig 1c and 1g.

Fig.1c. shows length of a straight line between the pharynx to primary branch. Fig. 1g indicates length of a line connecting the pharynx to primary branch along the luminal surface. Because the luminal surface is upheaved by cartilage at E18.5, Fig. 1g shows longer than 1c at E18.5, reflecting 3D morphology. We described quantification methods in materials and methods.

2-Non-proliferative luminal enlargement:

o Phase 2 is also proliferative, although in a reduced manner (x1.31). The authors should change the title to “low-proliferative” or eliminate any mention to proliferation on it.

This is important comment. We revised these in text.

o The authors should demonstrate that proliferation increase after induction with tamoxifen in *Nkx2.1CreERT2*; *LSL-KrasG12D* mice (Fig 2f), and quantify this proliferation.

Thank you for the suggestion. As we showed Reviewers 1 and 2, we quantified and added these data to Supplementary Fig. 3.

3-*Wnt5a-Ror2* signalling extends the tube:

o The authors already show *Wnt5* and *Ror2* is expressed only in the stromal compartment (figures 1c and 1h). It was interesting to confirm that conditional deletion of *Wnt5a* from epithelium gave a similar phenotype that controls (Fig 3d), but the conditional deletion of *Ror2* from epithelium (Fig 3i) is not relevant, and does not provide any crucial information. I would suggest relocating to supplemental or eliminating.

According to this suggestion, we relocated the *SHH^{Cre}; Ror2^{flox/flox}* to Supplementary Figure 5g, h.

o The authors should better characterize the tube diameter increase (phase 2 of tubulogenesis) in *Wnt5a^{-/-}* and *Foxg1Cre; Ror2flox/flox* mice: tube diameter, proliferation, apical enlargement and emergence...

We agree. Tube diameter and proliferation of *Wnt5a^{-/-}* and *Foxg1^{Cre}; Ror2^{flox/flox}* were quantified. We added these data into Supplementary Fig. 4a, b and 16a, b and mentioned in the manuscript. We further examined epithelial reshaping of *Wnt5a* at the phase 2. Although epithelial tissue structure is slightly altered in *Wnt5a^{-/-}*, apical enlargement was detected. By contrast, apical emergence was not observed in *Wnt5a^{-/-}*. Given that tracheal shape become shortened at phase1 tubulogenesis, epithelial structure might be indirectly affected by early elongation defects of *Wnt5a^{-/-}*. We added these data into Supplementary Fig. 16d, e and mentioned in the text.

o Minor point: change E14.5 to E11.5 when talking about Fig 3c in the text

We revised.

4-Radial polarization of subepithelial cells arrange SM morphology:

o Authors do not consider defects in cell intercalation as potential mechanism controlling tracheal morphogenesis despite their results support this hypothesis, which would also be backed by evidence in bibliography of the regulation by Wnt5a/Ror2 of the elongation of another tubular organ, the intestine, in mice during development through PCP (Cervantes et al., 2009, Yamada et al., 2010)

This is a very important point and sorry for omitting these papers. As we showed in Fig. 5, the Wnt5-Ror2 dependent radial cell polarity control the tube length of the trachea and esophagus, but neither stomach nor gut. Instead, as the reviewer commented, epithelial cell intercalation mediated by Wnt5a-Ror signaling is crucial for elongation of the stomach, duodenum and intestine. In this context, Wnt5a-Ror signaling regulates the PCP for proper epithelial cell intercalation (Cervantes et al., 2009, Yamada et al., 2010). In agreement with this distinct regulation, we found that Ror2 was actually expressed in epithelial cells of the stomach and gut but not in those of the trachea and esophagus. We added these data to Supplementary Fig 10. Thus different mechanisms play roles for tube elongation of trachea/esophagus and stomach/gut. We mentioned this point in discussion.

o Fig 5a should be properly quantified

We measured intensity of pMLC staining and confirmed that the level of pMLC intensity was significantly decreased.

However, we retracted Figure 5a because we concluded that pMLC in SM have nothing to regulate tracheal elongation. As the answer to reviewer2 comment, we newly generated SM-null mice, *Foxg1^{Cre}*, *Sr^{flox/flox}*. In these mutant mice, the elongation of the trachea was intact even though pMLC2 was lost, suggesting that pMLC2 is not involved in tracheal elongation.

o According to Fig 5g, Ror1/Ror2 KO tracheas show a significant difference in external diameter. This result shows that Ror1 might indeed have an interesting role in the control of tracheal morphogenesis (which is in contradiction with authors statements about Fig 3e-g results) and that both receptor functions seem to be redundant. Authors should analyse tubulogenesis phase 2 in these mice.

As reviewer mentioned, double knockout of Ror1/2 showed severe phenotype including short and expanded trachea compared to Ror2 knockout at E12.5, suggesting the Ror1 plays redundant role on phase1 tubulogenesis. We corrected manuscript. On the other hands, we could not find the morphological difference between Ror2 cKO and

Ror1/2 cKO at E18.5. These data indicates that Ror rector family is not involved in phase2 tubulogenesis. We added these data to Supplementary Fig. 16b, c.

5-Cartilage growth expands tube diameter:

o The authors should also analyze the SM cells, not only epithelial cells

Thank you for the suggestion. We examined SM tissue of the no-cartilage trachea at E16.5, see Supplementary Fig. 11, and confirmed the morphologies are indistinguishable.

DISCUSSION

6- The statement “in this study, we did not detect any changes in expression or distribution of cell-cell adherent complexes, cytoskeleton or cell polarity-related molecules in epithelial cell during cell reshaping” is too bold and further experiments would be necessary to address this issue. A confocal image (Fig S7) with different markers is not sufficient to demonstrate this point, which is clearly relevant.

We agree. To avoid misleading, we revised the description as below,

“In this study, we also failed to detect a change in expression or distribution of cell-cell adherent complexes, cytoskeleton or cell polarity-related molecules in epithelial cell during cell reshaping (Supplementary Fig. 15). However, it is still possible that endogenous epithelial factors are involved in trachea tubulogenesis. These factors should be explored in the future study.”

REVIEWERS' COMMENTS:

Reviewer #1 (Remarks to the Author):

I thank the authors for a mostly thorough response to my critiques.

However, the response to comment 1 and 2 were difficult to follow that was not helped by providing a figure with no legend. I believe though that I was able to decipher the figures and they support my previous concerns with the way the authors were interpreting their morphogenesis data.

1) I do not agree with the authors' rebuttal to point 1. The first part of the manuscript is concerned with understanding whether the expansion of the surface area can be accounted for by the expansion of epithelial cell numbers over the course of development. So this is a presentation about rates of change and it is a simple mathematical concept that this refers to slopes of curves, so to present ratios between each time point is just wrong.

Indeed, the flaw in this approach is beautifully demonstrated by the nice figure the authors provided for this reviewer that plotted slopes of epithelial expansion (reviewer figure top right). As is clearly demonstrated here (with remarkable stats on the curve fitting), the epithelial surface area is undergoing an accelerating expansion that is consistent with their conclusion. However, when they discuss this part by referring to "fold changes" they actually start to make wrong descriptions of the result, specifically:

"The luminal-surface area increased 28.2-fold from E12.5 to E18.5 (Fig. 1i upper). It increased 4.41-fold from E12.5 to E14.5, slow downed to increase 2.23-fold from E14.5 to E16.5, and then reaccelerated to increase 2.87-fold from E16.5 to E18.5 (Fig. 1i lower)."

In fact the actual data as presented in the reviewer panel shows that the rate of expansion of the epithelial surface area is simply accelerating, probably at a constant rate, throughout development. So in fact there is no "slowing down" and "reacceleration" as claimed by the authors. These are simply artifacts of using ratios at different time points to describe rates. So if the authors insist on continuing to present "fold changes" they will have to modify the text to avoid making the wrong description of the results.

In contrast to the acceleration of surface area, analysis of the rate of change of epithelial cell numbers shows an almost linear expansion (bottom row, left panel of reviewer figure) with some mild deceleration evident upon assessing slope change (reviewer panel bottom right) that can readily be explained by the reduction in proliferative rates. Comparing top right to bottom right makes a dramatic demonstration that cell numbers cannot account for surface expansion. As per the arguments above, describing these cell number expansions as fold changes in the context of this section is misleading. If fold-changes are to be presented then the description of the results has to be changed so they are not misrepresenting what the actual data is showing.

Finally, based on these results the conclusion of "biphasic morphogenesis" referred to at the bottom of page 5 can't be made at this point in the ms, because the observations likely just reflect the accelerating nature of rate of surface area expansion, rather than a truly biphasic process. The authors should modify their conclusions at this stage. However, the subsequent work in the manuscript demonstrates that to maintain this acceleration requires the PCP responses as worked out by the authors, so I don't think this necessarily diminishes the overall impact of the paper.

Point 2) The authors seem to have simply estimated cell numbers between 16.5 and 18.5, when in fact I was asking for them to actually measure surface area and cell numbers between 15.5 to 18.5 to try and define the transition phase in late stage surface expansion. However, I don't think this is such a critical point any more given that the slope analysis simply shows a pretty much

constant acceleration throughout development and no evidence of distinct phases of surface expansion.

--

Reviewer #2 (Remarks to the Author):

I am satisfied with the revisions. The authors have added additional data and revised the manuscript to address the majority of my concerns. Overall this is an excellent paper provides an extensive genetic and quantitative cell biology analysis supporting a novel model of trachea morphogenesis.

--

Reviewer #3 (Remarks to the Author):

The authors have performed experiments to address this reviewer's concern.

--

Reviewer #4 (Remarks to the Author):

In this reviewed version of the manuscript "Synchronized mesenchymal cell polarization and differentiation shape the trachea and esophagus" the authors show that different mechanisms regulating the length and diameter of the murine trachea. Their data seem to fit in a model in which Wnt-mediated SM polarized migration and ring cartilage development drive organ tubulogenesis.

The authors have addressed most of the questions/issues raised by this reviewer, and now the message of the article is clearly stronger and better supported. In summary, they have found that Ror2 is expressed in epithelial cells of the stomach and gut but not in those of the trachea and esophagus. Thus different mechanisms play roles for tube elongation of trachea/esophagus and stomach/gut, which is PCP dependent cell intercalation. And also that pMLC2 is not involved in tracheal elongation. I like this article very much. Although there are still some deficiencies, for instance a detail characterization of the molecular mechanisms is still missing, the article is now ready to be published.

The findings provided by the authors in this study are certainly interesting, novel and of high quality, and thus, are interesting for broad readers of Nature Communications. I'd like to congratulate the authors.